

# Estimation of turbulence dissipation rate and its variability from sonic anemometer and wind Doppler lidar during the XPIA field campaign

Nicola Bodini[1], Julie K. Lundquist[1,2], and Rob K. Newsom[3]

[1]Department of Atmospheric and Oceanic Sciences, University of Colorado Boulder, Boulder, Colorado, USA
[2]National Renewable Energy Laboratory, Golden, Colorado, USA
[3]Pacific Northwest National Laboratory, Richland, Washington, USA

*Correspondence to:* Nicola Bodini (nicola.bodini@colorado.edu)

**Abstract.** Despite turbulence being a fundamental transport process in the boundary layer, the capability of current numerical models to represent it is undermined by the limits of the adopted assumptions, notably that of local equilibrium. Here we leverage the potential of extensive observations in determining the variability of turbulence dissipation rate ($\epsilon$). These observations can provide insights towards the understanding of the scales at which the major assumption of local equilibrium between

generation and dissipation of turbulence is invalid. Typically, observations of $\epsilon$ require time- and labor-intensive measurements from sonic and/or hot-wire anemometers. We explore the capability of wind Doppler lidars to provide measurements of $\epsilon$. We refine and extend an existing method to accommodate different atmospheric stability conditions. To validate our approach, we estimate $\epsilon$ from four wind Doppler lidars during the 3-month XPIA campaign at the Boulder Atmospheric Observatory (Colorado), and we assess the uncertainty of the proposed method by data inter-comparison with sonic anemometer measurements

of $\epsilon$. Our analysis of this extensive dataset provides understanding of the climatology of turbulence dissipation over the course of the campaign. Further, the variability of $\epsilon$ with atmospheric stability, height, and wind speed is also assessed. Finally, we present how $\epsilon$ increases as nocturnal turbulence is generated during low-level jet events.

## 1    Introduction

Turbulence within the atmospheric boundary layer is critically important to transfer heat, momentum and moisture between the

surface and the upper atmosphere (Sobel and Neelin, 2006). Hence, global and regional models need an accurate representation of turbulence to produce precise atmospheric predictions of winds, temperature and moisture in the boundary layer. An accurate forecasting of these quantities has a critical impact on a variety of socio-economic activities, such as pollutant dispersion and air quality forecasting (Huang et al., 2013) and forest fires prediction and management (Coen et al., 2013). Wind energy production is also highly affected by turbulence in the boundary layer, as a lower power is generated when turbulence intensity

is high (Wharton and Lundquist, 2012), and turbulence also reduces the lifetime of wind turbines (Kelley et al., 2006).

     The production of turbulence kinetic energy in the boundary layer mainly takes place at large scales (Tennekes and Lumley, 1972). These large eddies then decay in smaller and smaller eddies through a "turbulence energy cascade" in the inertial sub-





range (Kolmogorov, 1941), until the length scales are small enough that the molecular diffusion is capable to dissipate the kinetic energy into heat in the viscous sub-range. Current models assume that the generation of turbulence within a grid cell (local production) is balanced by the dissipation $\epsilon$ of turbulence kinetic energy in the same grid cell (local dissipation). This assumption of local equilibrium is appropriate for stationary and homogeneous flow (Albertson et al., 1997), and therefore

it can be applied at coarse scales, with resolutions of the order of $3\mathrm{km}$ or larger. However, at scales of $\sim 1\mathrm{km}$ or finer, the fundamental assumptions of turbulence closures are broken (Nakanishi and Niino, 2006; Hong and Dudhia, 2012). Therefore, when using modes at fine horizontal resolution, the assumption of local equilibrium between generation and dissipation of turbulence is not valid anymore: turbulence produced in one grid cell can be advected downwind before being dissipated.

Hence, improved turbulence parametrizations are crucially needed to refine the accuracy of model results at fine horizon-
tal scales. Yang et al. (2017) showed that, when testing the turbine-height wind speed sensitivity to different parameters in the Mellor–Yamada–Nakanishi–Niino (MYNN) planetary boundary-layer scheme (Nakanishi and Niino, 2009) and the MM5 surface-layer scheme (Grell et al., 1994) of the Weather Research and Forecasting model (Skamarock et al., 2005) in a complex terrain region, roughly half of the wind speed variance was due to the accuracy of the parametrization of the turbulence dissipation rate. $\epsilon$ also controls the evolution of several boundary layer processes, such as cyclone formation and dissipation (Zhang
et al., 2009), the formation of frontal structures (Chapman and Browning, 2001; Piper and Lundquist, 2004), and the flow in urban areas and other canopies (Baik and Kim, 1999; Lundquist and Chan, 2007). Moreover, dissipation in aircraft vortices has a primary importance in aviation meteorology and air-traffic control (Gerz et al., 2005). Therefore, a correct representation of $\epsilon$ would improve the quality of numerical weather prediction. However, in order to improve turbulence parameterizations, the spatio-temporal variability of $\epsilon$ in the boundary layer needs to be studied in detail, as well as the dependence of $\epsilon$ with
atmospheric stability, orography, and turbulence itself, in order to understand at what spatio-temporal scale local imbalance becomes important.

Estimates of turbulence dissipation rate have been calculated from sonic anemometers on meteorological towers in the past (Champagne et al., 1977; Muñoz-Esparza et al., 2017) and hot-wire anemometers suspended on tethered lifting systems (Frehlich et al., 2006; Lundquist and Bariteau, 2014) with the inertial sub-range energy spectrum method (Oncley et al., 1996)
and the second-order structure function method (Frehlich and Sharman, 2004). Wind profiling radars have also been used to estimate $\epsilon$ (McCaffrey et al., 2017a), with the spectral width method. Wind Doppler lidars can also provide an extensive network of measurements of $\epsilon$ at different locations and at heights which are not accessible to traditional mast measurements. Four main methods are currently known to derive $\epsilon$ from lidar measurements, depending on the lidar scanning mode and measurement frequency: width of the Doppler spectra (Smalikho, 1995; Banakh et al., 1995), line-of-sight velocity spectrum (Banakh et al.,
1995; Drobinski et al., 2000; O'Connor et al., 2010), line-of-sight velocity longitudinal structure function (Frehlich, 1994; Banakh and Smalikho, 1997; Smalikho et al., 2005), and line-of-sight velocity azimuthal structure function (Banakh et al., 1996; Frehlich et al., 2006).

In this study, we prove the capability of wind Doppler lidars to provide precise estimates of $\epsilon$ by refining the approach proposed in O'Connor et al. (2010) to estimate $\epsilon$ from lidar line-of-sight velocity spectra, we assess its uncertainty, and present an
extensive analysis of the variability of $\epsilon$ in the atmospheric boundary layer. We estimate turbulence dissipation rate from the





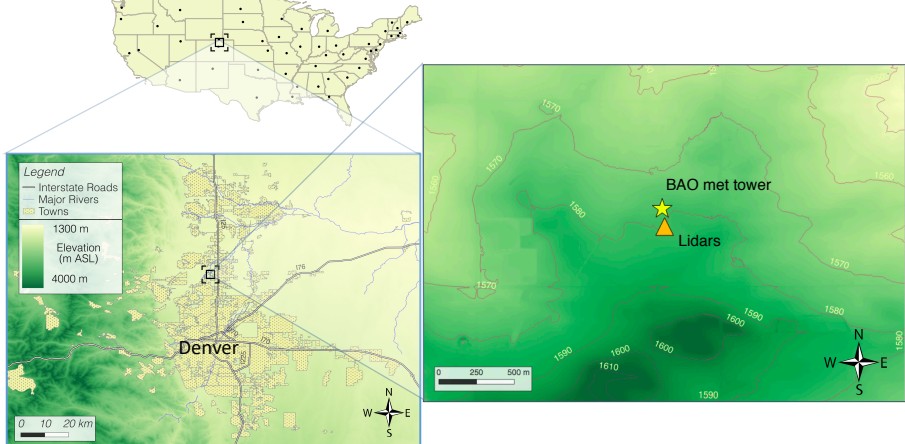

**Figure 1.** Map of the topography of the region where the XPIA field campaign took place.

3-month period of the eXperimental Planetary boundary layer Instrumentation Assessment (XPIA) field campaign (Lundquist et al., 2017), described in Section 2, from sonic anemometers and vertical profiling lidars, with the approach summarized in Section 3. The refinement of the method to derive $\epsilon$ from lidar to accommodate different stability conditions, and the quantification of its uncertainty are presented in Section 4. In Section 5 we assess the variability of $\epsilon$ with atmospheric stability, wind speed, and height, thus creating a climatology of turbulence dissipation. We finally focus, as case study, on how turbulence dissipation rate varies during nocturnal low-level jet events.

## 2 Data

To analyze the variability of turbulence dissipation rate, we use data from the meteorological tower and wind Doppler lidars deployed during the XPIA field campaign, summarized in Lundquist et al. (2017). The XPIA campaign, which took place at the Boulder Atmospheric Observatory (BAO) in northern Colorado between 2 March and 31 May 2015, was designed to explore the capabilities of multiple instruments to characterize different flow conditions in the boundary layer. As shown in the map in Figure 1, the region of the XPIA campaign is characterized by relatively flat terrain, with a few gentle hills south of the meteorological tower. The average elevation of the area is 1,584 MSL. Grass and low-crops fields surround the observatory, with some scattered trees and compact buildings.

### 2.1 Meteorological tower measurements

During XPIA, the 300-m BAO meteorological tower (Kaimal and Gaynor, 1983) had two 3D sonic anemometers (Campbell CSAT3) at each of six levels (50, 100, 150, 200, 250, and 300 m AGL), providing measurements with a frequency of 20Hz. The measurement resolution was generally less than $1 \cdot 10^{-3} \mathrm{m\,s^{-1}}$ in the horizontal and $5 \cdot 10^{-4} \mathrm{m\,s^{-1}}$ in the vertical. At each level, the two sonic anemometers were mounted pointing northwest ($334°$) and southeast ($154°$). In order to avoid tower





wake effects, data from the northwest sonics are discarded when the wind direction was between $111°$ and $197°$, while wind directions between $299°$ and $20°$ exclude data recorded by the southeast sonic (McCaffrey et al., 2017b). Data have been tilt-corrected according to the planar fit method described in Wilczak et al. (2001). An additional 5-m AGL tower was located 200m south-west of the BAO tower, and provided near-surface turbulent measurements. The sonic anemometer at this location operated with a frequency of 10Hz.

We quantify atmospheric stability from the 5-m tower data in terms of the Obukhov length $L$, defined as:

$$L = -\frac{\overline{\theta_v} \cdot u_*^3}{k \cdot g \cdot \overline{w'\theta_v'}} \tag{1}$$

where $\theta_v$ is the virtual potential temperature (K), calculated from the sonic anemometer virtual temperature data $T_v$ and the measured pressure $p$ as $\theta_v = T_v \left(\frac{p_0}{p}\right)^{R/c_p}$ with $p_0 = 1000$ hPa, and $R/c_p \approx 0.286$; $k = 0.4$ is the von Kármán constant; $g = 9.81$ m s$^{-2}$ is the gravity acceleration; $u_* = (\overline{u'w'}^2 + \overline{v'w'}^2)^{1/4}$ is the friction velocity (m s$^{-1}$); and $\overline{w'\theta_v'}$ is the kinematic sensible heat flux (Wm$^{-2}$). The turbulent quantities have been separated in average and fluctuating parts using the Reynolds decomposition with an averaging time of 30 minutes. This time scale is a common choice (De Franceschi and Zardi, 2003; Babić et al., 2012) when studying boundary layer processes, since it is generally longer than the turbulence time scales, but also shorter than the mean flow unsteadiness time-scales. As to atmospheric stability, we classify neutral conditions for $L \leq -500$m and $L > 500$m; unstable conditions for $-500$m $< L \leq 0$m; and stable conditions for $0$m $< L \leq 500$m (Muñoz-Esparza et al., 2012). Neutral conditions were rarely detected (less than 5% of the times) during the period of the campaign.

At the base of the BAO tower, a tipping-bucket rain gauge was used to measure precipitation. We have excluded from our analysis the times within one hour from precipitation events ($\sim 8\%$ of the times), as during these cases the measurement accuracy of both sonic anemometers and wind Doppler lidars drops.

## 2.2 Wind Doppler lidar measurements

Several vertical profiling and scanning wind Doppler lidars were deployed at XPIA. In this study, we focus on three vertical profiling lidars and one scanning lidar mainly used in vertical staring mode. All these instruments were co-located approximately 100m south of the BAO tower (Figure 1).

A WINDCUBE version 2 (v2) profiling lidar was deployed from 12 March to 8 June 2015. This lidar samples line-of-sight velocity in four cardinal directions with a nominal $28°$ zenith angle, followed by a fifth vertical beam. Range gates were centered on 40, 50, 60, 80, 100, 120, 140, 150, 160, 180, and 200m AGL. The retrieval of the actual wind speed from this measurement approach assumes horizontal homogeneity across the cone defined by the laser beams during the $\sim 4$s required to complete a sequence of measurements across the five beams.

Two WINDCUBE version 1 (v1) profiling lidars (Aitken et al., 2012; Rhodes and Lundquist, 2013) were deployed by the University of Colorado Boulder and the National Center for Atmospheric Research from 1 and 4 March 2015. These instruments measure line-of-sight velocity in four cardinal directions (nominal $28°$ zenith angle), with a range resolution of 20m, from 40 to 220 m AGL. The assumption of horizontal homogeneity of the flow in the sampling volume is again necessary



**Table 1.** Main technical specifications of the lidars at XPIA used in this study.

|  | WINDCUBE v2 | WINDCUBE v1 | Halo Streamline |
|---|---|---|---|
| Wavelength | $1.54\,\mu m$ | $1.54\,\mu m$ | $1.548\,\mu m$ |
| Receiver bandwidth | $\pm 57.5\,\mathrm{MHz}$ | $\pm 55\,\mathrm{MHz}$ | $\pm 25\,\mathrm{MHz}$ |
| Nyquist velocity ($B$) | $\pm 44\,\mathrm{m\,s^{-1}}$ | $\pm 42.3\,\mathrm{m\,s^{-1}}$ | $\pm 19.4\,\mathrm{m\,s^{-1}}$ |
| Signal spectral width ($\Delta\nu$) | $2.65\,\mathrm{m\,s^{-1}}$ | $3.39\,\mathrm{m\,s^{-1}}$ | $1.5\,\mathrm{m\,s^{-1}}$ |
| Pulses averaged ($n$) | 20000 | 10000 | 20000 |
| Points per range gate ($M$) | 32 | 25 | 10 |
| Range-gate resolution | $10-20\,\mathrm{m}$ | $20\,\mathrm{m}$ | $30\,\mathrm{m}$ |
| Minimum range gate | $40\,\mathrm{m}$ | $40\,\mathrm{m}$ | $15\,\mathrm{m}$ |
| Number of range gates | 11 | 10 | 200 |

to retrieve the actual wind vector. These instruments will be identified in the remainder of the analysis with their serial numbers, 61 and 68.

Finally, a Halo Photonics Streamline Doppler scanning lidar (Pearson et al., 2009) from the U.S. Department of Energy Office of Science Atmospheric Radiation Measurement program was deployed from 6 March to 16 April 2015. This lidar used

a range gate resolution of 30m, with 200 total range gates. However, the maximum range gate with an acceptable number ($> 30\%$) of valid measurements (SNR $> -20$dB) was at about 800m AGL. This scanning lidar was mainly used in a vertical staring mode. The scan strategy also included a 40-s plan-position-indicator (PPI) scan at an elevation angle of $60°$ once every 12min (from which the derivation of the horizontal wind speed is possible), a 10-min tower stare once per hour, and a target sector scan once per day to confirm heading relative to the tower (Newsom et al., 2017).

Table 1 includes the main technical characteristics of the three commercial lidar models considered in our analysis.

## 3 Methods to estimate turbulence dissipation rate $\epsilon$

### 3.1 Turbulence dissipation from sonic anemometer

Sonic anemometers data can be used to calculate turbulence dissipation rate with two different methods: the inertial sub-range energy spectra method and the second-order structure function method. Muñoz-Esparza et al. (2017) analyzed data at XPIA

and showed that the second-order structure function method has a lower error in estimating $\epsilon$ compared to the inertial sub-range energy spectra method, even when shorter overlapping temporal sub-windows are used to obtain a more regular pattern in the spectra. Therefore, we also apply the second-order structure function method to estimate $\epsilon$ from sonic anemometer measurements every 30s, for the 3-month period of XPIA. As already mentioned, data were excluded for wind directions waked by the tower . When neither of the two anemometers was affected by tower wakes, $\epsilon$ is defined as the average between the two

independent values obtained from the two sonics at each height.



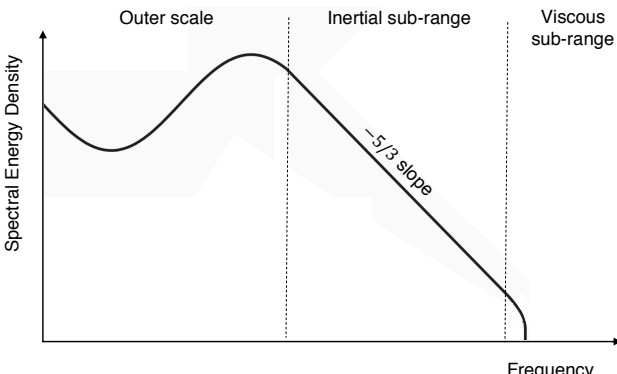

**Figure 2.** Turbulence energy spectrum according to Kolmogorov's hypothesis.

According to Kolmogorov's hypothesis, within the inertial sub-range the velocity increments, expressed as second-order structure function $D_U$ of the horizontal velocity $U$, can be related to $\epsilon$ as:

$$D_U(r) \equiv\; < [U(x+r) - U(x)]^2 > = \frac{1}{a}\epsilon^{2/3}r^{2/3} \tag{2}$$

where $< \cdot >$ denotes an ensemble average, and $a$ is the Kolmogorov constant. We assume $a = 0.52$, which is consistent with
the range of values present in the literature (Paquin and Pond, 1971; Sreenivasan, 1995). The spatial separations $r$, which has
to lie within the inertial sub-range, can be expressed as temporal velocity increments by invoking Taylor's frozen turbulence
hypothesis (Taylor, 1935), so that $\epsilon$ can be determined as:

$$\epsilon = \frac{1}{U\tau}\left[aD_U(\tau)\right]^{3/2} \tag{3}$$

where $D_U(\tau)$ is the second-order structure function of the horizontal velocity $U$ calculated over temporal increments $\tau$. For
every $\epsilon$ calculation (i.e. every 30s), the second-order structure function was calculated with a 2-min window for $\tau$, centered at
the nominal time at which $\epsilon$ is calculated. Then, the fitting to the theoretical model only used the time range between $\tau = 0.1$s
and $\tau = 2$s. Such a short temporal separation in the data is expected to lie well within the inertial sub-range, therefore excluding
the undesired contributions from the outer scales which would undermine Kolmogorov's fundamental assumptions. Moreover,
despite the reduced size of the chosen time range, the high temporal resolution of the sonic anemometers still guarantees an
adequate number of data points to allow a robust estimation of the structure function. Data inspection confirms that the desired
theoretical $\tau^{2/3}$ slope is observed in the chosen range for $\tau$ (example shown in the Supplement).

## 3.2   Dissipation from Doppler lidar

Wind Doppler lidars can provide a great improvement of our understanding of the variability of turbulence dissipation thanks
to the ease of their deployment in different locations and the long measurement range allowed by several commercial models.
To do so, robust methods to estimate $\epsilon$ with lidars are necessary, and their uncertainty has to be assessed. For this purpose, we




follow and refine the novel approach described in O'Connor et al. (2010) to estimate $\epsilon$ from vertical profiling lidars or scanning lidars used in vertical staring mode. For homogeneous and isotropic turbulence, within the inertial sub-range, the turbulent energy spectrum (Figure 2) can be expressed according to the Kolmogorov (1941) hypothesis in terms of wavenumber $k$ as:

$$S(k) = a\epsilon^{2/3}k^{-5/3} \tag{4}$$

where $a \simeq 0.52$ is the one-dimensional Kolmogorov constant. The wavenumber $k$ can be written in terms of a length scale $L = 2\pi/k$ by invoking Taylor's frozen turbulence hypothesis (Taylor, 1935). By integrating (4) over the wavenumber space, the variance $\sigma_v^2$ of the de-trended observed line-of-sight velocity can be obtained:

$$\sigma_v^2 = \int\limits_k^{k_1} S(k)dk = -\frac{3}{2}a\epsilon^{2/3}\left(k_1^{-2/3} - k^{-2/3}\right) = \tag{5}$$

$$= \frac{3a}{2}\left(\frac{\epsilon}{2\pi}\right)^{2/3}\left(L_N^{2/3} - L_1^{2/3}\right) \tag{6}$$

and therefore if the length scales are properly chosen (and consistent with how $\sigma_v$ is computed) then $\epsilon$ can be calculated without the need of systematically computing turbulence energy spectra. In (6), the length scale $L_1$ for a single sample interval is given by:

$$L_1 = Ut + 2z\sin\left(\frac{\theta}{2}\right) \tag{7}$$

where $U$ is the horizontal wind speed, $t$ is the dwell time, $\theta$ the half-angle divergence of the lidar beam, and $z$ the height
AGL. Since Doppler lidars generally have a very small $\theta$, the second term in (7) is typically negligible. For $N$ samples, the length scale becomes $L_N = NUt$. The horizontal wind speed $U$ can be derived from the line-of-sight velocity measurements performed by the profiling lidars, with the assumption of horizontal homogeneity of the flow over the probed volume. In the case of the Halo Streamline, no information about the horizontal wind can be derived from the measurements in the vertical staring mode, which only measures the vertical component of the wind speed. $U$ is then retrieved from a sine-wave fitting from
the VAD scans that are performed every 12min. The heights at which the measurements are taken during the tilted VAD scans are not the same as the heights sampled in the vertical staring mode. Therefore, for each considered level in the vertical staring mode, $U$ is determined from a linear interpolation of the wind speed retrieved at the two closest heights during the VAD scans. Considerations about the error introduced by this procedure on the estimation of $\epsilon$ will be discussed in Section 4.

Lidar measurements are inherently affected by signal noise as well as possible variations of the aerosol fall speeds, which
provide additional contributions to the observed variance. Therefore, the variance $\sigma_v^2$ in (6) can be written as the sum of three different terms, which can be considered to be independent of one other (Doviak et al., 1993):

$$\sigma_v^2 = \sigma_w^2 + \sigma_e^2 + \sigma_d^2 \tag{8}$$

$\sigma_w^2$ is the desired net contribution from atmospheric turbulence, from which the estimation of $\epsilon$ can be made. The additional contributions to the variance are due to the instrumental noise ($\sigma_e^2$) and the variation in the aerosol terminal fall speeds within



the measurement volume from different sample intervals ($\sigma_d^2$), which however can safely be neglected since the particle fall speed is typically $< 1\,\mathrm{cm\,s^{-1}}$. For a heterodyne Doppler lidar, Pearson et al. (2009) provides the following expression for the noise contribution to the variance, as a function of the signal-to-noise ratio (SNR):

$$\sigma_e^2 = \frac{\Delta\nu^2\sqrt{8}}{\alpha N_p}\left(1+\frac{\alpha}{\sqrt{2\pi}}\right)^2 \tag{9}$$

where $N_p$ is the accumulated photon count:

$$N_p = \mathrm{SNR}\,nM \tag{10}$$

In this expression, $n$ is the number of lidar pulses which are averaged to get a profile, and $M$ is the number of points sampled within a single range gate to get a velocity estimate. $\alpha$ is the ratio of the lidar photon count to the speckle count (Rye, 1979):

$$\alpha = \frac{\mathrm{SNR}}{\sqrt{2\pi}}\frac{B}{\Delta\nu} \tag{11}$$

where $B$ is the bandwidth, equivalent to twice the Nyquist velocity, and $\Delta\nu$ is the signal spectral width.

The noise contribution to the observed variance determines an additional area below the turbulence spectrum in its high-frequency region (Frehlich, 2001) which, if not removed, would induce an overestimation of $\epsilon$. Therefore, the turbulence dissipation rate can be estimated as:

$$\epsilon = 2\pi\left(\frac{2}{3a}\right)^{3/2}\left(\frac{\sigma_v^2 - \sigma_e^2}{L_N^{2/3} - L_1^{2/3}}\right)^{3/2} \tag{12}$$

This method relies on the assumption that both length scales $L_1$ and $L_N$ are within the inertial sub-range. Therefore, the choice of the number of samples $N$ to use should be carefully addressed, since only the turbulence contributions in the inertial sub-range should be included in the calculation. We discuss in detail this choice and its relationship with different atmospheric stability conditions and heights in the next section.

## 4   Error in turbulence dissipation rate estimates from lidar measurements

Although promising, the method to calculate $\epsilon$ from lidar data presented in the last section needs to be carefully analyzed in relation to its fundamental assumptions and its uncertainty, especially given the limited temporal resolution of lidar measurements. In this section we refine the method to derive $\epsilon$ from lidar data by discussing, in relationship with different atmospheric stability conditions and heights, the choice of the number of samples $N$ to use for the calculation of the variance of the de-trended line-of-sight velocity and corresponding length scales. Moreover, we assess the uncertainty of this method by systematically comparing $\epsilon$ values from lidar measurements with what is obtained from the sonic anemometers, and we discuss how the estimation error changes with height in the boundary layer.

While the high temporal resolution of sonic anemometers facilitates the identification of sizable samples within the inertial sub-range, for lidars, the length of the samples used to estimate the variance of the line-of-sight velocity should be accurately





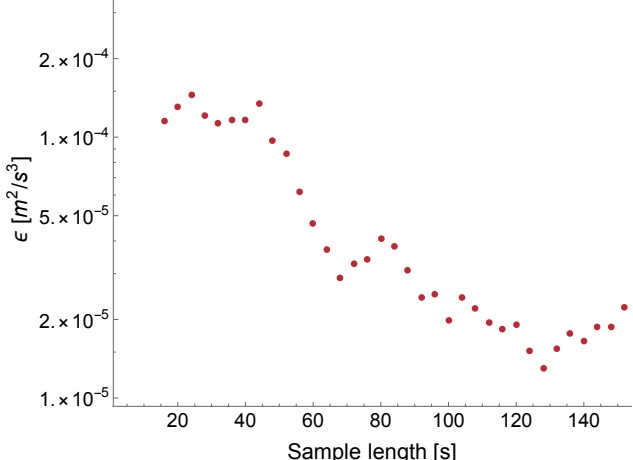

**Figure 3.** Example of the dependence of $\epsilon$ on the sample length used in the calculation. Data from the WINDCUBE v2 lidar at 100m AGL, 30 March 2015, 14:20 UTC.

chosen. In fact, the shorter the sampling time, the higher the measurement error in the estimate of the variance of line-of-sight velocity would be, because of the higher relative contribution of the instrumental noise. According to the formulation in Lenschow et al. (2000), the measurement error $\Delta\sigma_w^2$ in the turbulence contribution to the observed variance $\sigma_w^2$ can be estimated as:

$$5 \quad \Delta\sigma_w^2 \simeq \sigma_w^2 \sqrt{\frac{4\sigma_e^2}{N\sigma_w^2}} \qquad (13)$$

so it therefore decreases as the number of samples $N$ increases, with the hypothesis that the noise contribution $\sigma_e^2$ to the variance of each velocity sample used to estimate $\epsilon$ is similar to the ensemble mean error.

On the other hand, if the sampling time is too long, the variance will incorporate undesired contributions from the large-scale processes, which would cause a severe underestimation of $\epsilon$. Figure 3 shows how the estimated value of $\epsilon$ varies with the

10 sample length used in the calculation, for a case using the WINDCUBE v2 data at 100m AGL. As long as the sample length stays within the inertial sub-range (up to $\sim 50$s in the case shown), $\epsilon$ stays approximately constant. However, the estimate of $\epsilon$ decreases by up to an order of magnitude when the contributions from the outer scales are erroneously included in the calculation, which uses expressions that are valid strictly only within the inertial sub-range.

Moreover, since different atmospheric stability conditions are inherently characterized by different turbulence scales (Kaimal

15 et al., 1972), the transition from the inertial sub-range to the outer scales occurs for different sample lengths, depending on the atmospheric stability. Figure 4 shows examples of turbulence spectra calculated over 15-min intervals for data measured by the WINDCUBE v2 lidar at 100m AGL in different stability conditions. For stable conditions (panel a), the transition from the inertial sub-range (which can be identified by comparing the slope of the spectrum with the theoretical $-5/3$ value shown by the dashed line) to the outer scales occurs at a higher frequency compared to the unstable case (panel b). Therefore, the choice



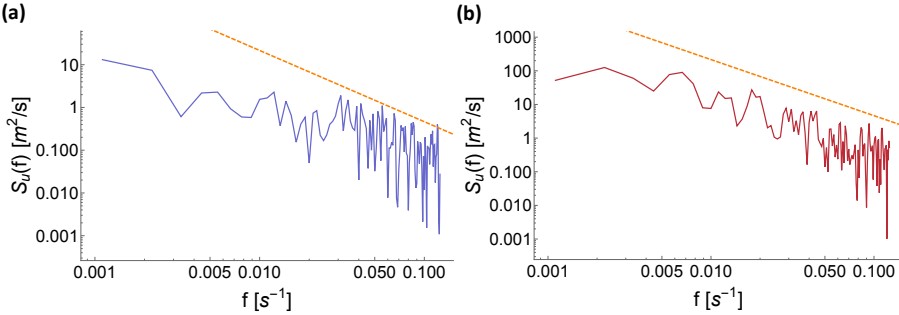

**Figure 4.** Turbulence energy spectrum for a stable case (panel a - 2 April 2015, 03:00 UTC), and an unstable case (panel b - 3 April 2015, 22:15 UTC, calculated from 15 minutes of data measured by the WINDCUBE v2 at 100m AGL. The dashed lines represent the theoretical $-5/3$ slope expected in the inertial sub-range.

of the number of samples $N$ to use in the calculation should change accordingly. As a general rule, we expect shorter time scales to be adequate for stable conditions, when the turbulent eddies in the boundary layer are smaller, while longer scales would be more suitable during unstable conditions, characterized by larger convective eddies that can be fully captured only when using larger scales. Moreover, different altitudes can also impact the extension of the inertial sub-range, with a wider

development expected at higher heights.

To estimate the appropriate time scales which best balance these competing factors, we calculate $\epsilon$, at each height from each of the considered lidars, using several values for the number of samples $N$ used in the calculation. At the heights where there is correspondence between lidar and sonic anemometer measurements, we then compare the $\epsilon$ values from the lidars with the corresponding $\epsilon$ calculated at the meteorological tower. The estimates of $\epsilon$ from sonic anemometers and lidars have

been calculated at slightly different time stamps, given the unavoidable difference in the nominal measurement time stamps of instruments operating with different temporal resolutions. Given the inherent turbulent nature of $\epsilon$ and its remarkable range of variability, the comparison between the time series from sonic anemometers and lidars could be flawed by the effect of the turbulent high-frequency variability of $\epsilon$. Moreover, since this analysis is focused on the assessment of the appropriate time scales for different stability conditions, consistency with the time scale used to calculate turbulent fluxes for the determination

of the Obukhov Length $L$ is advisable. Therefore, a 30-min running mean is applied to the time series of $\epsilon$ from both sonic anemometers and lidars before comparing the estimates from the different instruments.

The result of this comparison is reported in Figure 5, which shows how the median absolute error (MAE) between sonic and lidar estimates of $\epsilon$ varies with the time scale (calculated as $Nt$, where $t$ is the dwell time of the considered lidar) used to estimate $\epsilon$ for the WINDCUBE v2 lidar, for different atmospheric stability conditions, at 100m AGL. As the used sample

length increases, the average error in $\epsilon$ estimated from lidar initially decreases from the high values related to the strong noise contribution at short time scales. Then, a minimum in the error is reached. As the size of the sample further increases, the average error rises again, due to the incorporation of undesired contributions from the outer scales. Moreover, as expected, the minimum error for stable (and neutral) conditions is found for shorter time scales compared to unstable conditions. Also, the





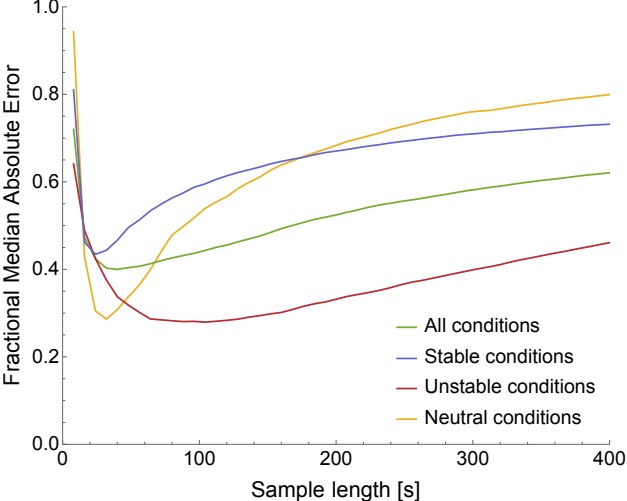

**Figure 5.** Median absolute error between $\epsilon$ estimates from sonic anemometer and WINDCUBE v2 lidar data at 100m AGL during the whole period of the XPIA campaign, as a function of the sample length used to estimate $\epsilon$ from lidar data.

minimum error in stable conditions is higher than minimum error for unstable conditions, since the need of using a shorter time scale implies a higher relative contribution of the instrumental noise to the error. The same qualitative pattern is found for all the considered lidars, at all heights. At each height, for each lidar and for each stability classification, we select the time scale that produces the lowest median absolute error compared to the sonic anemometer estimates of $\epsilon$: this can be
interpreted as the longest time scale that does not include substantial contributions from the undesired outer scales. The visual identification of the inertial sub-range in turbulence energy spectra from lidar measurements in different stability conditions confirms the magnitude of the selected scales, and can therefore be considered as an alternative way to assess the appropriate sample sizes in field campaigns with no available co-located tower measurements. Table 2 summarizes the selected time scales for the considered lidars for the different stability conditions (neutral conditions are not shown because rarely occurred, with
a frequency lower than 5%), as well as the average from all the instruments, at 100m AGL. As expected, the larger eddies which characterize unstable conditions determine the need for a longer time scale to capture the influence of all the scales included in the inertial sub-range, while for stable conditions a shorter time scale is more appropriate. The median error is higher during stable conditions (average: MAE = 51%) compared to unstable conditions (average: MAE = 29%), as expected and as observed in other studies (Smalikho and Banakh, 2017).
Looking at the variability of the results with height, we find that the optimal time scales increase with height. At those heights $< 300$m AGL where lidar measurements do not match the level of any sonic anemometer on the meteorological tower, the adopted time scales are chosen as averages between the scales at the closest levels covered by sonics. For the Halo Streamline lidar, whose measurements are considered up to 800m AGL in this study, we determine the appropriate sample sizes by linearly extrapolating aloft, for each stability condition, the sequence of the chosen scales at the lower levels, where
a comparison with the meteorological tower data is possible. The linear trend matches well the observed results up to 300m,





**Table 2.** Time scales which minimize the median absolute error (MAE) in the comparison between $\epsilon$ from sonic anemometers and lidars at 100m AGL for stable and unstable conditions. Results for neutral conditions are not shown since these were rarely detected during the campaign.

|  | Stable conditions | | Unstable conditions | |
| --- | --- | --- | --- | --- |
|  | Time scale | MAE | Time scale | MAE |
| WINDCUBE v2 | 24s | 44% | 88s | 27% |
| WINDCUBE v1 - 61 | 24s | 49% | 96s | 28% |
| WINDCUBE v1 - 68 | 32s | 49% | 72s | 26% |
| Halo Streamline | 28s | 62% | 73s | 37% |
| Average | 27s | 51% | 82s | 29% |

with $R^2 > 0.9$ for all stability conditions (plot shown in the Supplementary Materials). Moreover, the rationality of the chosen scales at high altitudes has been confirmed after inspecting the extension of the inertial sub-range in turbulence spectra from the Halo Streamline lidar data (figure not shown).

Once the appropriate time scales have been identified at each height, considerations about how the error in lidar estimates of $\epsilon$ varies with height can be made. Figure 6 shows how the median absolute error between lidar and sonic estimates of $\epsilon$ changes with height, for all the levels at which sonic anemometers were mounted on the BAO tower. When a match between the height of lidar measurements and the level of the sonics was not present, the median error shown in the plot has been estimated as the average between the errors at the two closest lidar range gates. For the WINDCUBE v1-68, data at 50m AGL are not available because of measurement contamination due hard strikes with the guy wires of the meteorological tower. The same issue invalidates measurements at 140m AGL from the WINDCUBE v1-61; therefore for this lidar the comparison with the sonic anemometer at 150m AGL shown in this plot has been performed using only the lidar data measured at 160m AGL. For the Halo Streamline, measurements below 105m AGL show a high percentage of low SNR data and therefore are not reported. For the WINDCUBE lidars, the median absolute error slightly increases with height, likely because of the severe reduction of the number acceptable measurements at higher levels, and it always stays below 50%. For the Halo Streamline lidar, the median error stays almost constant in the considered portion of the boundary layer. It is reasonable to explain the higher error ($\sim +10\%$) of the Halo Streamline compared to the WINDUBE lidars at 100m AGL as a consequence of the necessary approximations adopted in the determination of the horizontal velocity $U$ for the Halo Streamline lidar, as explained in Section 3.2. However, the magnitude of this additional error due to the reduced frequency in determining $U$ for the Halo Streamline is comparable with the additional uncertainty related to the drop of instrumental performance that the WINDCUBE show at higher levels. Therefore, the estimates of $\epsilon$ from the Halo Streamline can be considered physically robust in the lowest few hundred meters of the boundary layer.

Possible sources for the discrepancy found between $\epsilon$ from sonic anemometers and lidars might arise from the different temporal resolution and sampling volumes of the various instruments, as well as the 100m spatial separation between the lidar





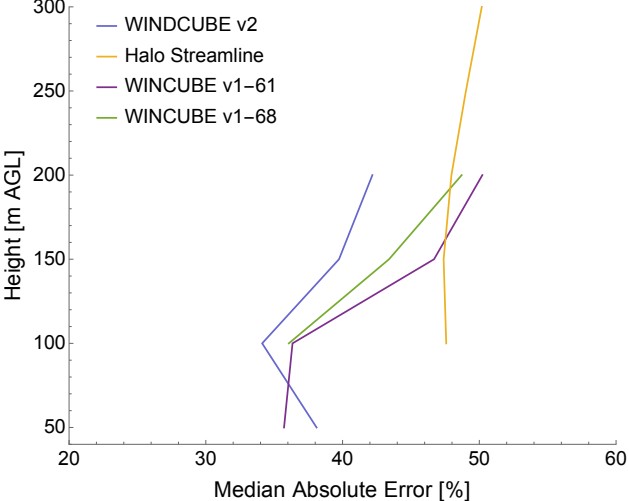

**Figure 6.** Varibiality of the median absolute error (calculated from all atmospheric stability conditions) between lidar and sonic anemometer estimates of $\epsilon$ with height, for the four considered lidars.

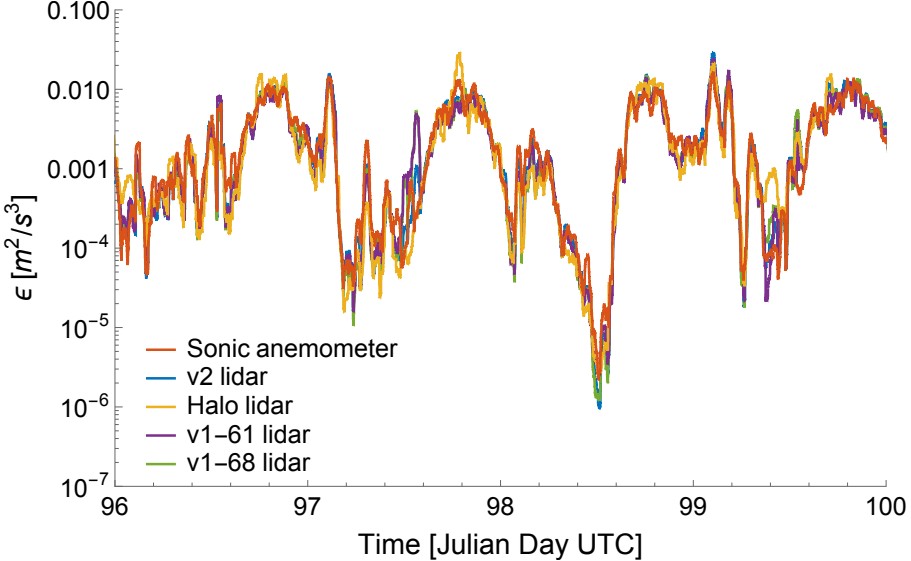

**Figure 7.** Time series from 6 April 2015 00 UTC to 10 April 2015 00 UTC comparing $\epsilon$ from sonic anemometers and all the considered lidars at 100m AGL. Data have been smoothed with a 30-min running mean.

site and the BAO meteorological tower. In any case, given the wide range of variability of $\epsilon$, which can span $\sim 6$ orders of magnitude during its typical diurnal cycle (Section 5), the obtained magnitudes of the error prove that the refined method to retrieve $\epsilon$ from lidar measurements gives robust estimates of turbulence dissipation rate. The accommodation for different



stability conditions in the choice of the time scales used in the method considerably reduces, especially for stable conditions, the magnitude of the errors (obtained through propagation of errors) found in the original study (O'Connor et al., 2010). To visualize the good agreement between sonic anemometer and lidar estimates of $\epsilon$, Figure 7 shows the time series for a portion of the XPIA campaign, with values from all the considered instruments at 100m AGL. A clear diurnal pattern is revealed, with

higher values of turbulence dissipation during the day, and differences of several orders of magnitude between daytime and nighttime values of $\epsilon$. These results will be explored in more detail in Section 5. A systematic comparison between $\epsilon$ estimates from sonic anemometers and the WINDCUBE v2 lidar at 100m AGL is shown by the density histograms in Figure 8, for the whole period of the XPIA campaign, for different stability conditions. The coefficient of determinations $R^2$ are also reported in the plots. The good agreement between data from sonic anemometer and lidars is confirmed, with unstable conditions showing

a better performance ($R^2 = 0.89$ for the smoothed time series) compared to stable conditions ($R^2 = 0.74$). Moreover, the plots show the effect of the choice of applying the 30-min running mean before comparing $\epsilon$ values from the different instruments. In the figure, the panels on the left compare $\epsilon$ without any temporal filter, while the panels on the right show the comparison between time series after the 30-min running mean has been applied. The application of the 30-min running mean to the $\epsilon$ time series increases the correlation between the different time series. In any case, even for the raw time series, the values of the

coefficient of determination are always greater than 0.6. Also, the application of this filter does not considerably change the choice of the appropriate time scales.

## 5 Variability of turbulence dissipation rate

Once the capability of the method to provide accurate estimates of $\epsilon$ from lidar data has been tested, the variability of turbulence in the boundary layer can be assessed, using data from the various instruments deployed at XPIA.

The time series of $\epsilon$ shown in the previous Section revealed that, during the course of the day, $\epsilon$ changes by several orders of magnitude. To better explore this diurnal variability, Figure 9 shows the daily climatology of turbulence dissipation rate, calculated as median of the data from the sonic anemometer, WINDCUBE v2 lidar and Halo Streamline lidar. Plots for the two WINDCUBE v1 lidars are shown in the Supplementary Materials, as similar to the results from the WINDCUBE v2. A general good agreement between the climatology from sonic anemometers and lidars can be observed. A definite diurnal pattern

is evident from each panel. As expected, the mainly quiescent conditions at night determine low values of turbulence dissipation rate ($\epsilon \sim 10^{-5} - 10^{-4}\mathrm{m^2s^{-3}}$), while daytime convection increases the median turbulence dissipation in the boundary layer by several orders of magnitude ($\epsilon \sim 10^{-2}\mathrm{m^2s^{-3}}$). During nighttime, however, the median values of $\epsilon$ show more variability than during daytime conditions, as traces of intermittend bursts of $\epsilon$ can be detected in the climatology. We will investigate these changes in $\epsilon$ in more detail, by relating the variability of $\epsilon$ with wind speed, especially in the case of nocturnal low-level jets.

Also, the study of the climatology of $\epsilon$ can give insights on how $\epsilon$ changes with height. The analysis of the climatology from the sonic anemometers (Figure 9 a), which allow measurements of $\epsilon$ at 5m AGL, shows how $\epsilon$ is higher close to the surface throughout the day, while above 50m AGL the change of $\epsilon$ with height is less noticeable. A similar result can be found from lidars, which provide $\epsilon$ measurements starting at 40m AGL for the WINDCUBE v2, and 75m AGL for the Halo





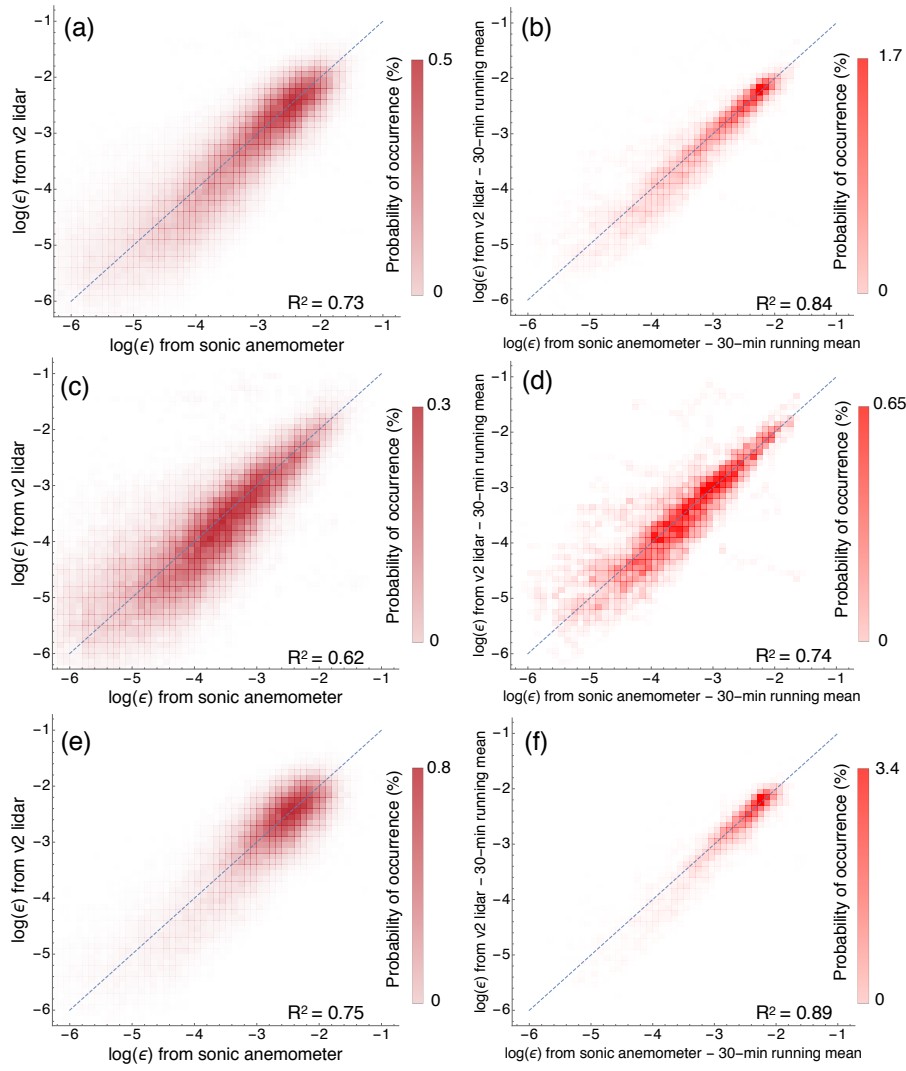

**Figure 8.** Correlation between $\epsilon$ values from sonic anemometer and WINDCUBE v2 lidar at 100m AGL for the whole period of the XPIA campaign, using the selected time scales for the estimation of $\epsilon$ from lidar data. The color scales represent the probability of occurrence in percentage, and the dark dashed lines show perfect correlation. (a) All stability conditions, raw data (MAE = 62%); (b) all stability conditions, 30-min running mean applied (MAE = 34%); (c) stable conditions, raw data (MAE = 67%); (d) stable conditions, 30-min running mean applied (MAE = 44%); (e) unstable conditions, raw data (MAE = 58%); (f) unstable conditions, 30-min running mean applied (MAE = 27%).

Streamline, with reduced variability of $\epsilon$ with height in the majority of the sampled height range. The slight increase of $\epsilon$ above $\sim 600$m AGL at night for the Halo Streamline lidar (Figure 9 c) can be explained due to the higher frequency of good-quality measurements at higher levels during high wind speed events, which determine higher turbulence, as will be shown





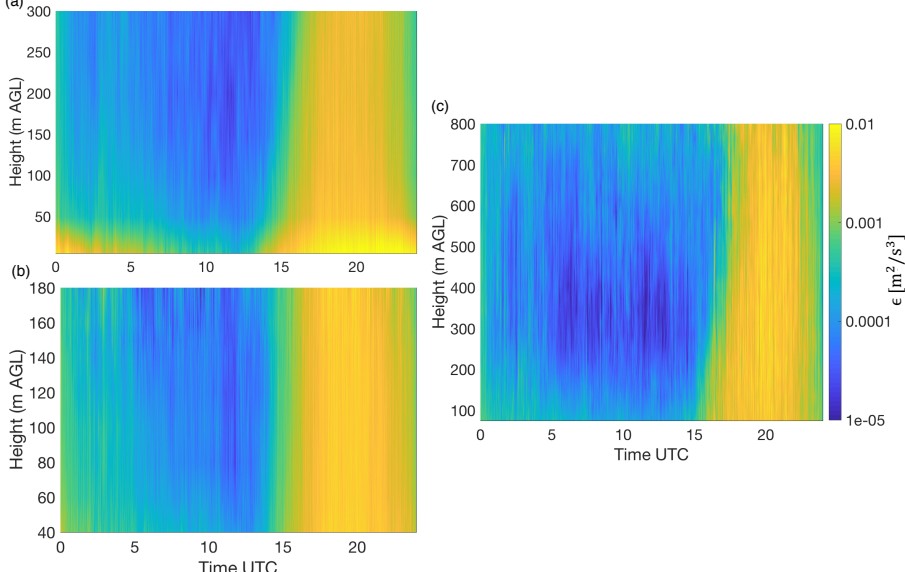

**Figure 9.** Daily climatology of turbulence dissipation rate derived from sonic anemometer data (panel a), WINDCUBE v2 lidar (panel b), and Halo Streamline lidar (panel c). Results from the two WINDCUBE v1s are included in the Supplementary Material.

later in this Section. A systematic analysis of how turbulence dissipation rate varies with height is shown in Figure 10. For each instrument, the percentage difference in $\epsilon$ is shown, and it is calculated by taking as reference value the closest in time value of $\epsilon$ determined from the sonic anemometer at 5m AGL, so that a common reference level is identified for all the instruments. The continuous line in the plot shows the median value at each height, while the shaded band represents the 1st and 3rd

quartiles of the data distribution. The plot confirms that turbulence dissipation rate shows most of its variability with height close to the surface. A 75% decrease in the median $\epsilon$ value is observed moving from 5m AGL to 50m AGL for the sonic anemometer data. An additional increase of height determines a lower rate of average reduction of $\epsilon$ with height, with the median $\epsilon$ values for the sonics experiencing an additional 15% reduction (compared to the reference 5m AGL level) between 50m AGL and 300m AGL. Variations of comparable magnitude are also found for the lidar data, for both the WINDCUBE v2

and the Halo Streamline. In any case, the spread around the median value is quite extensive at all the considered heights for all the instruments.

     The effect of different atmospheric stability conditions on turbulence dissipation can be investigated in more detail by relating $\epsilon$ with the correspondent Obukhov length ($L$) values, which is used here as a measurement of stability. Figure 11 shows the relationship between turbulence dissipation rate and the absolute value of $L$, for all instruments, at 100m AGL. For

each instrument, we sort $\epsilon$ based on $L$. Then, we sub-divide the $\epsilon$ data in correspondence of equally-spaced (in the logarithmic space) $L$ bins. The median $\epsilon$ in each group is shown by the continuous line in the plot. The shaded area shows the range between the 1st and 3rd quartiles. Results from raw $\epsilon$ data (i.e. without the application of the 30-min running mean) are shown in the plot. However, no substantial differences arise from the use of the smoothed time series. Different stability conditions systematically





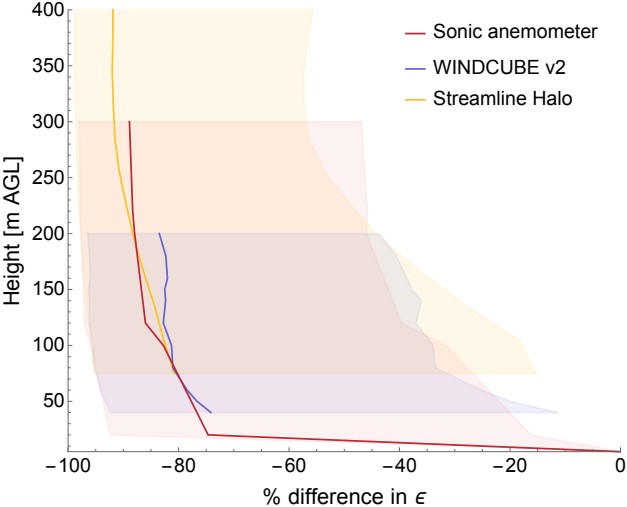

**Figure 10.** Turbulence dissipation rate as a function of height for different instruments. The variability with height is expressed as percentage change assuming as reference level 5m AGL. The continuous line in the plot represents the median value for different instruments, while the shaded area creates a band corresponding to the 1st and 3rd quartiles of the values.

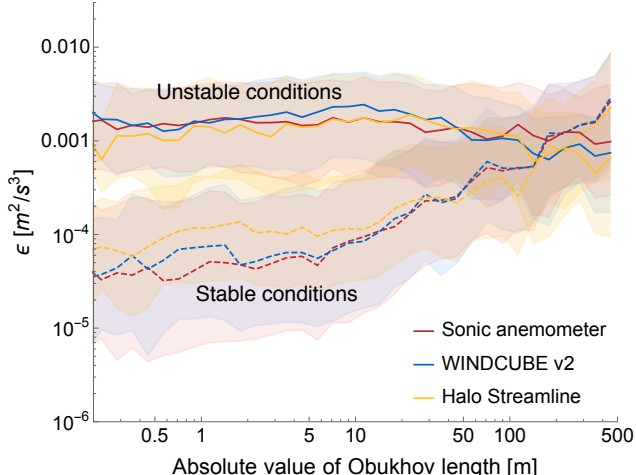

**Figure 11.** Turbulence dissipation rate (measurements at $100$m AGL) as a function of the absolute value of the Obukhov Length $L$. The thick lines in the plot represent the median value for the different instruments, while the shaded area creates a band corresponding to the 1st and 3rd quartiles of the distributions. Continuous (dashed) lines for unstable (stable) conditions.

change the magnitude of turbulence dissipation rate, with median $\epsilon$ values during strong stable conditions ($L \rightarrow 0^+$m) generally two orders of magnitude lower than what is found for strongly unstable conditions ($L \rightarrow 0^-$m). Moreover, as the atmospheric stability conditions become less strong, with an increase in the absolute value of $L$, the median $\epsilon$ values tend to converge to a





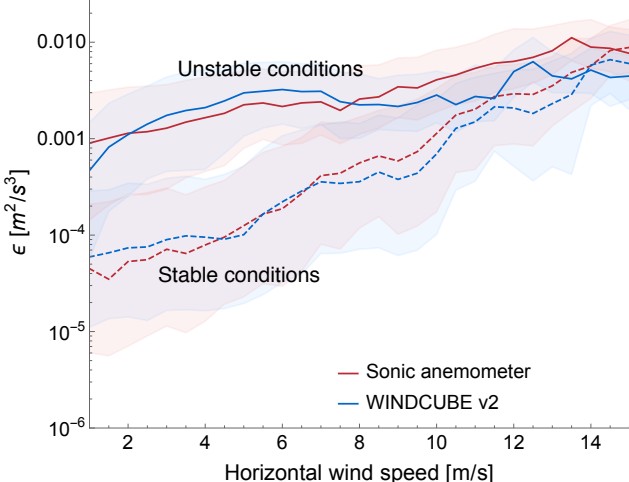

**Figure 12.** Turbulence dissipation rate as a function of the 2-min average wind speed, as measured at 100m AGL. The thick lines in the plot represent the median value for the different instruments, while the shaded area creates a band corresponding to the 1st and 3rd quartiles of the distributions. Continuous (dashed) lines for unstable (stable) conditions.

common value, with $\epsilon$ in stable conditions recording a higher increase compared to the change in $\epsilon$ for different values of $L$ in unstable conditions. Results from neutral conditions $|L| > 500$m are not shown as they rarely occurred at the site during the field campaign.

Different wind speed regimes can also have a strong impact of on the development and subsequent dissipation of turbulence.
Figure 12 relates turbulence dissipation rate with 2-min average wind speed, for different stability conditions, at 100m AGL. The same sampling technique described for Figure 11 to define median $\epsilon$ values, shown by the continuous line, has been applied in this case. Data from the Halo Streamline are not included here since the reduced temporal availability of horizontal wind speed measurements (once every 12min) does not guarantee a precise estimation of the variability of $\epsilon$ with wind speed for this instrument. For both the sonic anemometer and the WINDCUBE v2 lidar data, a strong dependence of $\epsilon$ on wind speed can be
observed. As wind speed increases, more turbulence is generated - and therefore dissipated - in the boundary layer. The median $\epsilon$ increases of about one order of magnitude as wind speed intensifies from $1\,\mathrm{m\,s^{-1}}$ to $15\,\mathrm{m\,s^{-1}}$. This positive correlated trend is found for both stable and unstable conditions, with $\epsilon$ in stable conditions being more subject to variations with wind speed compared to $\epsilon$ in unstable conditions. Also, the difference in $\epsilon$ during distinct stability conditions becomes less pronounced as the wind speed increases. Therefore, high wind speeds seem to determine strong turbulence - and turbulence dissipation -
without any significant dependence on the stability condition.

## 5.1 Turbulence dissipation rate during nocturnal low-level jet events

The accurate numerical representation of nocturnal low-level jets has a crucial importance. In fact, this sudden increase of wind speed aloft at night has been shown to have a primary effect on turbulent transport (Prabha et al., 2007), clear-air turbulence



(Banta et al., 2002), storm formation (Curtis and Panofsky, 1958), and forest fire propagation (Barad, 1961) and wind resources (Vanderwende et al., 2015). In all these cases, turbulence represents an essential driving mechanism, and therefore turbulence dissipation needs to be represented with particular attention. During XPIA, nocturnal low-level jets have been observed several times (Lundquist et al., 2017). As case study, Figure 13 shows how wind speed, wind direction, and turbulence dissipation

rate varied during the night between 6 - 7 April 2015, as measured by the Halo Streamline lidar. The analysis of the weather maps for this period reveals no frontal passage (confirmed by the absence of any significant shifts in wind direction during the considered period, as shown in Figure 13 b); no precipitation was recorded; and the analysis of ceilometer data reveals clear sky. A considerable increase in wind speed (up to $14\,\mathrm{m\,s^{-1}}$, Figure 13 a) can be observed between 21 and 23 LT. In correspondence to this jet, turbulence dissipation rate (Figure 13 c) increases by at least an order of magnitude throughout the

considered vertical portion of the boundary layer, reaching values of $\sim 10^{-2}\mathrm{m^2 s^{-3}}$ which are comparable to what is observed during daytime convection, as can be seen between 15 and 17 LT in the presented case. This abrupt increase of $\epsilon$, which interrupts the normal decrease of $\epsilon$ due to the transition from daytime convection to nocturnal quiescence, can also clearly be detected in the time series shown in Figure 7. After the end of the low-level jet event, the return to quiescent conditions, typical of the nighttime stable boundary layer, causes a considerable reduction of turbulence dissipation rate. Therefore, the turbulence

generated by the strong wind acceleration during nocturnal low-level jets can deeply modify the daytime climatology of $\epsilon$, determining the temporary increases which have been detected in the analysis of the climatology in Figure 9.

## 6   Conclusions

Turbulence parametrizations currently used in numerical models have been proved to have considerable limitations which undermine the quality of representations of processes in the atmospheric boundary layer. A crucial parameter in this regard is

the turbulence dissipation rate ($\epsilon$). Currently, most mesoscale planetary boundary layer models make the assumption of local equilibrium between production and dissipation of turbulence, without however having a full understanding of the scales and associated atmospheric conditions which break this hypothesis. In this study, we have demonstrated the value of observations from both in situ and remote sensing instruments in providing insights on the variability of turbulence dissipation rate, and we have assessed how $\epsilon$ changes with atmospheric stability, wind speed, and height in the boundary layer.

Besides using traditional approaches to estimate $\epsilon$ from sonic anemometers, we have refined the novel approach proposed by O'Connor et al. (2010) which enables the use of wind Doppler lidars to survey the variability of $\epsilon$. Our analysis provides recommendations about the choice of the length of sample of lidar measurements to use to calculate $\epsilon$. In fact, the properties of the turbulence energy spectra for different atmospheric stability conditions have to be taken into account to balance the competing needs of keeping the sampled scales within the inertial sub-range, while minimizing the impact of the instrumental

noise. Longer time scales are appropriate for unstable conditions, while shorter scales should be used in stable cases. Also, the choice of the appropriate sample size should take into account the variability of turbulence spectra with height, with longer scales more suitable aloft. The choice of the appropriate time scales can be made by either comparing lidar estimates of $\epsilon$ with



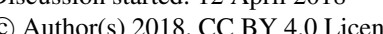

**Figure 13.** Nocturnal low-level jet case study. (a) Variability of wind speed from 20 UTC, 6 April 2015, to 8 UTC, 7 April 2015, as measured by the Halo Streamline lidar. (b) Wind direction at 116m AGL, during the same period of time. (c) Correspondent variability of turbulence dissipation rate $\epsilon$ as derived from the Halo Streamline measurements.



sonic anemometer data or inspecting the properties of the turbulence spectra from lidar measurements in different stability conditions and heights.

We have tested our methodological considerations by calculating $\epsilon$ from four Wind Doppler lidars deployed during the XPIA field campaign at the Boulder Atmospheric Observatory in Spring 2015, and we have systematically compared the lidar

estimates of $\epsilon$ with reference data from sonic anemometer measurements to determine the appropriate time scales to use in the calculation. Our results reveal a good agreement between lidar and sonic anemometer estimates of $\epsilon$, with median differences lower than $30\%$ in unstable conditions, and lower than $50\%$ in stable conditions. Given the range of variability of $\epsilon$, which spans several orders of magnitude throughout its diurnal cycle, and the $100\,\mathrm{m}$ horizontal separation between the lidar site and the meteorological tower, we consider these results satisfactory.

The promising results of this method make a considerable amount of data, measured in the recent years with vertical-profiling lidars, now potentially available to create an extensive database of turbulence dissipation rate for different atmospheric and topographic conditions. Wide deployments of lidars can in fact provide measurements in several different locations and at heights which are not accessible to traditional tower measurements. The analysis of the XPIA dataset reveals that different stability conditions have a considerable impact on determining the magnitude of $\epsilon$, with the dependence on the Obukhov length

$L$ revealing a clear trend. In stable conditions, $\epsilon$ increases as the atmosphere becomes less stratified, while $\epsilon$ decreases (but of a more reduced amount) in unstable conditions as the atmosphere tends to become more neutral. This dual pattern determines the diurnal climatology of $\epsilon$, with lower values during nighttime quiescent conditions and increased turbulence during the daytime convection, as would reasonably be expected. However, the general pattern of the climatology of $\epsilon$ strongly varies based on turbulence generation and dissipation due to the magnitude of wind speed. We have found that higher wind speeds

determine increased turbulence dissipation, with the gap between $\epsilon$ values in stable and unstable conditions becoming less pronounced as the wind speed increases. Therefore, important boundary layer processes such as nocturnal low-level jets can induce a substantial increase of $\epsilon$ at night, with values which can reach those of daytime convective turbulence. Finally, we have shown how most of the variability of $\epsilon$ occurs in the lowest part of the boundary layer, with a $75\%$ reduction from 5m AGL to 50m AGL.

The results from this dataset represent a significant progress towards the full understanding of how turbulence dissipation varies in the boundary layer. Future work will include testing the capability of lidars to measure turbulence dissipation rate in complex terrain, with potential case studies including mountain waves phenomena and diurnal circulations, as well as during other specific boundary layer processes, such as horizontal rolls (Brooks and Rogers, 1997). A complete database of the variability of $\epsilon$ in different terrains would in fact improve our understanding of the main drivers which determine

the development and dissipation of turbulence in various conditions. Once the variability of $\epsilon$ will be fully captured using different datasets, the scales at which the assumption of local equilibrium is broken will be assessed, and the implementation of improvements to the turbulence parametrizations used in numerical models will be possible.





*Data availability.* The data of the sonic anemometers and wind Doppler lidars at the XPIA field campaign are publicly available at https: //a2e.energy.gov/data.

*Competing interests.* The authors declare that they have no conflict of interest.

*Acknowledgements.* The authors thank Dr. Matthieu Boquet and Dr. Ludovic Thobois at Leosphere for providing the technical parameters
of the WINDCUBE v1 and v2 needed for this analysis. We also appreciate Dr. Boquet's and Mr. Evan Osler's (Renewable NRG) efforts to
provide the v2 to the XPIA field campaign. Support for JKL and NB is provided by the National Science Foundation (AGS-1554055) under
the CAREER program.



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
