# Peer review of "Estimation of turbulence dissipation rate and its variability from sonic anemometer and wind Doppler lidar during the XPIA field campaign"

_Atmospheric Measurement Techniques, 2018_

## Referee Comment (RC1) · Anonymous Referee #2 · 8 May 2018

In this manuscript, a technique to measure turbulence dissipation rate from Doppler lidar observations is presented using data collected from several Doppler lidars during XPIA. The dissipation rates are compared with those from sonic anemometers for verification (and to determine the sample length for the best agreement). Statistics of dissipation are presented for the experiment, which serve as a brief climatology of dissipation at the site.

The manuscript is generally written and organized well, and results in this manuscript are of significant interest to a wide audience in the Doppler lidar and boundary-layer fields. Still, there are some significant omissions in the description of the technique and

how the presented results are interpreted. As such, I recommend that this manuscript be reconsidered for publication after major revisions, after the following concerns have been addressed.

General/major comments:

a) How exactly is the turbulence dissipation calculated using the Doppler lidar data? More details need to be added to Sect. 3.2 so that this technique could be applied by a reader. From the Halo data, it must be the vertical staring observations. From the V1/V2 profiler data, which beam position is used (and why)? Was dissipation calculated from each beam separately, and the mean of those used?

While isotropy is assumed, turbulence is rarely isotropic in the boundary-layer, especially under stable conditions when turbulent eddies are more horizontally oriented. As such, there could be differences (particularly with the Halo which just uses vertical beam) between the lidar estimates and sonic anemometer estimates (which use the horizontal variance alone) from anisotropy. This should be briefly discussed.

b) In Sect. 4, the sampling length for calculation of dissipation during stable, neutral, and unstable conditions is chosen as the minimum of the MAE between the sonic and lidar estimate. This is fine when there is sonic anemometer data for both verification and classifying stability, but most sites that this technique could be applied to will not have coincident sonic measurements. How could this technique be applied to other sites, where the turbulence characteristics/stability might be quite different? This is a major limiting factor in the applicability of this technique, and currently there is no discussion of how this could be applied to other sites given this limitation. Also, does the minimum in the MAE vary between slightly stable and strongly stable conditions, when the inertial subrange may be much smaller? Should the analysis in Fig. 5 be done with more stability classifications (strongly stable/unstable, weakly stable/unstable, neutral)?

Perhaps this technique could be refined so that the sample length varies with the outer scale of the inertial subrange, as determined from the Doppler lidar data alone. Then,

the technique could be easily applied to other lidar data. Alternatively, the authors could add a short section (a few paragraphs) on how this technique could be applied at locations without sonic anemometer data for stability and determination of the sample length to use.

Specific comments:

a) p. 2 line 5; p. 21 line 27: Here, the authors make the case that both production and dissipation of TKE need to be known for turbulence closure. The authors state that by measuring dissipation, the scales at which the assumption of local equilibrium are broken will be assessed. However, in order to do this, production must also be measured. The authors should add a few statements on how production of TKE can be measured for the full closure.

b) Figure 1 caption: Would be good to clarify that contours in the right panel are in m.

c) p 3 line 1: Spell out XPIA in full here, for those unfamiliar with the project.

d) p. 4, line 2: Was this sonic also a CSAT3 or was it different? If it was a different type of sonic, are there differences in the design that may cause the observed dissipation to be much higher than for CSAT 3 (possibly more obstructions, if it's an RM Young anemometer) as later discussed in Sect. 5? Given its importance to the results, more details should be provided about this sonic, its siting, and any QC applied to it (was any data thrown out when it was waked by what it was mounted on)?

e) Table 1: Can the pulse width (FWHM) be added as a row to this table, as well? This will be useful in understanding the smallest eddies that can be resolved by a given lidar.

f) p. 5 line 19: How did the measured dissipation rates between the two sonic anemometers compare to each other when both were unwaked? Were they often similar, or were there often substantial differences? This might be useful to form a 'baseline' estimate of how much uncertainty is in any dissipation measurement from

the sonic anemometers themselves.

g) Eq. 5: By using this equation to estimate dissipation, it is implicitly assumed that the line-of-sight atmospheric variance ($\sigma w2$ in Eq. 8) is strictly the result of turbulent motion. However, non-turbulent motions such as gravity waves in a stable layer may increase the line-of-sight variance but are not turbulent, thus there is little dissipation with them. Under these conditions, turbulence dissipation would be overstated. This may be especially important at the BAO for westerly winds, due to the close presence of mountains to the west that may induce mountain waves when the atmospheric conditions permit. This may affect the statistics later presented in Sect. 5, as dissipation may be overestimated due to the presence of these waves.

h) Eq. 8: In the term $\sigma w2$ it should be clarified that this is not the true atmospheric variation of the wind, as the smallest scales of turbulence are not resolved by the lidar.

i) p. 8 line 23: Do the line-of-sight velocities need to be de-trended? Since the windows over which the variance is calculated is short (<1 min), the de-trending will effectively remove variance contributions from large eddies, especially during unstable conditions, causing an underestimate of variance (and consequently dissipation).

j) p. 9 line 5: Since the sampling window is so short, measurement uncertainty/representiveness (i.e., Lenschow et al 1994) is a significant factor in the quality/error of the variance measurement as well and should be mentioned.

k) Figure 4: Could vertical lines be added to denote the inertial subrange and/or sample length used?

l) p. 10 line 15: Could an equation be included here for how exactly the metric presented in Fig. 5 is calculated? I assume the error is normalized by some value (as the y-axis is unitless), but this is unclear. Without this information, it is difficult to interpret Fig. 5. The caption for Fig. 5 needs to be clarified accordingly, as well.

m) p. 14 line 33: As SNR typically decreases with range, is it possible that the increase
in dissipation above 600 m is due to more noisy/random errors in the line-of-sight mea-
surements above 600 m? Thus, the increase above this height is not real (due to
atmospheric turbulence), but instead due to increasing measurement errors.

n) Figure 9: The labels on these plots are small and difficult to read. Could they be
made larger?

o) p. 17 line 1: Are there other studies that also confirm the finding here that there is
a significant gradient in dissipation right near the ground, but the changes are much
smaller above 50 m? What physically results in this large almost order of magnitude
change in dissipation from the surface upwards? It would be good to expand on this.
Without justification or other studies that show similar results, these results seem a
little suspect. Was the 5-m sonic near anything that may obstruct the flow to cause
dissipation to be so large?

p) Sect 5.1: This LLJ event is atypical compared to most in the Great Plains,
where the LLJ slowly reaches a wind speed maxima in the middle of the night,
after which the wind speed slowly decreases. The rapid decrease in wind speed
at 03 UTC seems more like there was some other disturbance (possibly on
the mesoscale) that resulted in the jet diminishing. Looking at the tower data
(https://www.esrl.noaa.gov/psd/technology/bao/browser/), there was also about a 45
degree wind shift at the time the LLJ ended. Based on surface observation maps
(http://www2.mmm.ucar.edu/imagearchive1/surface/ict/20150407/sfc_ict_2015040703.gif,
http://www2.mmm.ucar.edu/imagearchive1/SatSfcComposite/20150407/sat_sfc_map_2015040704.gif)
, there was a Denver cyclone in the area with an associated quasi-stationary front
near the BAO site. Is it possible that the observed increase in dissipation was not from
the LLJ itself, but is induced by a front (possibly the quasi-stationary front drifting over
the site) or disturbance in the vicinity? Could a different LLJ event be chosen for this
analysis? Otherwise, the text must be modified accordingly to make it clear that this
observed behavior is not typical for LLJs and the presence of this quasi-stationary
front likely plays a role.

Technical corrections:

a) Figure 4 caption: Should be a ) after 22:15 UTC.

b) p. 12 line 26: WINDCUBE is misspelled.

References:

Lenschow, D. H., Mann, J., & Kristensen, L. (1994). How long is long enough when measuring fluxes and other turbulence statistics? J. Atmos. Ocean. Tech. Technol., 11, 661–673.
* * *

---

## Referee Comment (RC2) · Anonymous Referee #1 · 9 May 2018

**General Comments:**
This manuscript uses three months of data collected during the XPIA field campaign from 4 lidars and sonic anemometers on a 300m tower to calculate turbulence dissipation rates. The manuscript furthers a method of calculating epsilon from OConnor etal 2010, using the line of sight velocity spectra, and conducts an error analysis to determine the time scale that produces the most accurate turbulence dissipation rates. The minimum mean absolute error is determined for stable, unstable, and neutral conditions with good agreement with sonics, particularly when averaged to 30-minutes. These results are then compared for different heights, wind speeds, Obukhov length, and during one nocturnal low-level jet case to understand the variability of turbulence dissipation rates in a large range of conditions.

Overall, this is a well-conducted error analysis that allows for further analysis of variability of a difficult-to-measure quantity. My major concern comes from the fact that in situ observations are necessary for this analysis to be reasonably, so the extension of the method to a broad number of lidar sites without sonic anemometers seems unlikely.

I recommend publication of this manuscript after major revisions, as listed below.

**Specific Comments:**
The fact that on many plots the v1s are not included is worrisome; looks like only the good results are shown. A short sentence that results are similar is insufficient. It deserves a short discussion on the differences, benefits and drawbacks of each of the systems and why there is expected to be some variation (even if small).

These results of appropriate time scales for reducing error are very interesting, but can they be applied everywhere? With the dependence on stability and scales of turbulence, terrain would undoubtedly have a large affect on the time scales with minimum error. If there are no sonics available for the error analysis done here, and individual spectra need to be inspected to find an appropriate inertial range, the method breaks down and isn't reasonable. This problem needs to be addressed here.

- page 2, line 5: Is the 3km scale a result from Albertson et al also? If so, move citation to end of sentence. If not, include citation for this fact also
- page 5, table 1: include the temporal resolution of each lidar
- page 7, lines 14-15: what is theta here? How small is small?
- page 7: Are LOS velocity spectra calculated for all beams or only the vertically pointing? If all, isotropy must be assumed. Clarify and comment on this.
- page 10, lines 4-5: why is a wider inertial range expected at higher altitudes?
- page 14, line 15: this final sentence doesn't make sense. Why would the filter change the choice of shorter time scales being averaged?

**Technical Corrections:**
- On all figures with units of epsilon shown, use $m^2\ s^{-3}$, not $m^2/s^3$
- When referring to figure subplots, remove space between number and letter (check AMT standard)
- Include the time scale of epsilon on all plots
- Yellow lines are hard to see, especially the yellow shading. I appreciate the use of consistent colors for each instrument across all figures, but need a better choice for yellow. If v1s are used rarely, use purple or green instead of yellow, maybe the same color with different weights or dashes?

- line 7: accurate forecast
- 1: small enough that molecular diffusion is capable of dissipating

- 7: when using models
- 34: velocity spectra. We assess the uncertainty of this method, and present
- 5-6: as a case study… during a nocturnal low-level jet event
- 18: measurement accuracy or precision (not resolution)
- 14-15: For atmospheric stability, we classify neutral conditions as L … unstable conditions as … stable conditions as…
- 24: who deployed the v2?
- 30: wrong dates for the v1s
- 19: remove space after tower
- 5: which must be within
- 9: define $k_1$ and N earlier
- 6: period after equation
- 22-23: different heights and atmospheric conditions
- 11: looks more like 40s, not 50s
- 24: found to be at shorter time scales than unstable
- 9-10: because they occurred less than 5% of the time
- 9: due to hard strikes
- 10: v1-61, so the comparison… 150m AGL has been performed using only this lidar's data…
- 8 conditions and smoothing
- 12: note the time scales of the raw time series
- 20: lower case section (only capitalize when referring to Section X)
- 23: Materials, and are
- 26 & 27: space before units
- 28: intermittent
- 1: lowercase section
- 2: confusing wording
- 14: not all instruments
- 1-2: $L>0$ and $L<0$
- 4: impact on
- 11: increases of 1-2 orders (stable increases two orders)
- 1: wind energy resources
- 18: cite Yang et al, (2017). Sensitivity of turbine-height wind speeds to parameters in planetary boundary-layer and surface-layer schemes in the weather research and forecasting model. *Boundary-Layer Meteorology*, *162*(1), 117-142.

Figure 1: legend on right plot is not readable – match size of left legend. Colors on right subplot does not correspond to color scale legend on left subplot – include a new color scale for this subplot also

Table 1: WINDCUBE v1 (61 & 68)

Figure 5: Use a different color than yellow (purple?)

Figure 6: variability (misspelled); indicate which time scale is used for each stability class: minimum MAE for optimized Nt, at the appropriate time scales?

Figure 8: labels columns (raw and smoothed) and rows (all stability, stable, unstable) on figure

Figure 9: label instruments on figures; nighttime variability mentioned in text is hard to see on this color scale – maybe change to "jet" blue-red scale; I'd prefer the y-axes to be the same on the two left plots, or at least both start at 0; are these 30-minute or raw values?

---

## Author Response (AR1)

**RESPONSE TO REVIEWER #1**

*In this document, the reviewer's comments are in black, the authors' responses are in red.*

The authors thank the reviewer for their thoughtful and productive comments.

**General Comments:**

This manuscript uses three months of data collected during the XPIA field campaign from 4 lidars and sonic anemometers on a 300m tower to calculate turbulence dissipation rates. The manuscript furthers a method of calculating epsilon from O' Connor et al 2010, using the line of sight velocity spectra, and conducts an error analysis to determine the time scale that produces the most accurate turbulence dissipation rates. The minimum mean absolute error is determined for stable, unstable, and neutral conditions with good agreement with sonics, particularly when averaged to 30-minutes. These results are then compared for different heights, wind speeds, Obukhov length, and during one nocturnal low-level jet case to understand the variability of turbulence dissipation rates in a large range of conditions.

Overall, this is a well-conducted error analysis that allows for further analysis of variability of a difficult-to-measure quantity. My major concern comes from the fact that in situ observations are necessary for this analysis to be reasonably, so the extension of the method to a broad number of lidar sites without sonic anemometers seems unlikely.

Thank you for finding our work interesting and useful!

I recommend publication of this manuscript after major revisions, as listed below.

**Specific Comments:**

The fact that on many plots the v1s are not included is worrisome; looks like only the good results are shown. A short sentence that results are similar is insufficient. It deserves a short discussion on the differences, benefits and drawbacks of each of the systems and why there is expected to be some variation (even if small).

The results for the v1s are shown in Figures 6, 7, but not in Figures 9, 10, 11, 12. This choice was due to the fact that adding other lines and shaded areas to plots that are already really dense would have substantially increased the difficulty for a reader to read the plots, while not providing any new substantial knowledge as the results for the v1s are very similar to what obtained for the v2. However, all the plots for the v1s are included in the Supplementary Material (except for the correspondent of Figure 10 due to data contamination for the v1s at some heights for hard strikes).

Thank you for noticing that the fact that all the plots for the v1s can be found in the Supplement was not pointed out in a clear enough way in the manuscript.

In Section 5, the presentation of Figure 9 and its caption already included sentences to refer to the Supplementary Material for the plot for the WINDCUBE v1s. In the description of Figure 11 we have now included the following sentence: "The Supplementary Material includes the plot for the WINDCUBE v1s, which provide results very similar to what shown here". Also, the caption of the Figure now includes: "Results from the two WINDCUBE v1s are included in the Supplementary Material". In a similar way, the introduction of Figure 12 now includes the following: "(results for the WINDCUBE v1s are included in the Supplementary Material as very similar to what is found for the v2)". And the caption of the Figure now includes: "Results from the two WINDCUBE v1s are included in the Supplementary Material."

We expect to find the main variations in the results between the WINDCUBE lidars and the Halo lidar, given the difference in the scan pattern used (4,5 beams for the v1s, v2, while vertical stare for the Halo). To make this clear, we have included and modified the following sentences in Section 3.2:

"For the WINDCUBE lidars, the variance of the observed line-of-sight velocity $\sigma_v^2$ can be calculated as average from all the beams. In doing so, we include turbulence contributions from both the horizontal and vertical dimensions, and we make the limiting (Kaimal et al. 1972, Mann 1994) assumption of isotropic turbulence. For the Halo Streamline lidar, which operated in a vertical stare mode, $\sigma_v^2$ is calculated from the vertically pointing beam, and therefore $\epsilon$ will strictly include turbulence contributions only in the vertical dimension, thus possibly determining different values compared to what is retrieved from the WINDCUBE lidars.

Another difference due to the different scan patterns used by the considered lidars is related to the determination of the horizontal wind speed U. For the WINDCUBE lidars, U can be derived from the line-of-sight velocity measurements from the different beams, with the assumption of horizontal homogeneity of the flow over the probed volume. In the case of the Halo Streamline, no information about the horizontal wind can be derived from the measurements in the vertical staring mode, which only measures the vertical component of the wind speed. U is then retrieved from a sine-wave fitting from the VAD scans that are performed every 12 min".

Moreover, we have added and modified the following sentences in Section 4:

"It is reasonable to explain the higher error ($\sim$ +10%) of the Halo Streamline compared to the WINDCUBE lidars at 100m AGL as a consequence of the differences in the spatial dimensions that are samples by the two lidars. While the lidar beams of the WINDCUBE are tilted, and they therefore include turbulence contributions in the horizontal dimension (which is the only contribution considered in the determination of $\epsilon$ from the sonic anemometers), $\epsilon$ from the Halo Streamline is only retrieved using information from the vertically pointing beams. Moreover, the necessary approximations adopted in the determination of the horizontal velocity U for the Halo Streamline lidar, as explained in Section 3.2, likely determine an additional error increase for this lidar."

*References:*

- *Mann, J., 1994. The spatial structure of neutral atmospheric surface-layer turbulence. Journal of fluid mechanics, 273, pp.141-168.*
- *Kaimal, J.C., Wyngaard, J.C.J., Izumi, Y. and Coté, O.R., 1972. Spectral characteristics of surface-layer turbulence. Quarterly Journal of the Royal Meteorological Society, 98(417), pp.563-589.*

These results of appropriate time scales for reducing error are very interesting, but can they be applied everywhere? With the dependence on stability and scales of turbulence, terrain would undoubtedly have a large effect on the time scales with minimum error. If there are no sonics available for the error analysis done here, and individual spectra need to be inspected to find an appropriate inertial range, the method breaks down and isn't reasonable. This problem needs to be addressed here.

We have refined our approach to propose an alternative to use when measurements from co-located sonic anemometers are not available. We have included in the manuscript the following additional subsection:

**4.1 Determination of the optimal time scales to retrieve $\epsilon$ from lidars in absence of co-located sonic anemometers**

The availability of multiple sonic anemometers co-located with the lidars at XPIA has allowed for a direct comparison between $\epsilon$ estimates from different instruments to determine the optimal length scales, in different stability conditions, to use when retrieving $\epsilon$ from Doppler lidar measurements. This approach does not require the direct calculation of spectra from the line-of-sight velocity measured by the lidars, and therefore it represents a time-efficient technique. However, the proposed method is only viable when sonic anemometers are deployed in the near vicinity of a lidar, and when measures of atmospheric stability are available.

When a comparison with sonic anemometer data is not possible, the appropriate time scale to use in the lidar retrieval of $\epsilon$ can be determined by finding the maximum wavelength within the inertial sub-range in the velocity spectra from the lidar measurements. To do so, spectral models can be fitted to the observed spectra. Several models have been proposed for turbulence spectra in different stability conditions (Kaimal et al., 1972; Panofsky, 1978; Olesen et al., 1984). We test the spectral model proposed by Kristensen et al. (1989), which proposes expressions for both the cases of an isotropic and an anisotropic horizontally homogeneous flow. To validate our results and test this alternative approach to derive $\epsilon$ from lidar measurements, we use data from the Halo Streamline lidar to estimate the maximum wavelength $\lambda_z$ within the inertial subrange. Since the Halo mainly operated in a vertical stare mode during XPIA, we consider the following expression for the turbulence spectrum of the vertical component of the wind speed:

$$S(k) = \frac{\sigma_z^2 l_z}{2\pi} \frac{1 + \frac{8}{3}\left(\frac{l_z k}{a(\mu)}\right)^{2\mu}}{\left[1 + \left(\frac{l_z k}{a(\mu)}\right)^{2\mu}\right]^{5/(6\mu)+1}} \tag{16}$$

where $k$ is the wavenumber, $\sigma_z$ is the standard deviation of the vertical component of the wind speed used to compute the spectrum, $l_z$ is the integral scale of the vertical velocity along the horizontal flow trajectory, and the parameter $\mu$ controls the curvature of the spectrum. We use $\mu = 1.5$, which provides a good match with our experimental spectra, as also found in previous studies (Lothon et al., 2009; Tonttila et al., 2015). The parameter $a$ can be expressed as a function of $\mu$ as:

$$a(\mu) = \pi \frac{\mu \Gamma\left(\frac{5}{6\mu}\right)}{\Gamma\left(\frac{1}{2\mu}\right)\Gamma\left(\frac{1}{3\mu}\right)} \tag{17}$$

We calculate spectra using 10-min consecutive data, and we fit the spectral model to the experimental data, leaving out frequencies greater than 0.2Hz, which are affected by instrumental noise (Frehlich, 2001), not modeled here. An example of a measured spectrum and the fit resulting from the model are shown in Figure 9. The transition wavelength $\lambda_z$ between the

inertial sub-range and the outer scales can be expressed as a function of the integral scale $l_z$ and the parameter $\mu$:

$$\lambda_z = \left[ \frac{5}{3} \sqrt{\mu^2 + \frac{6}{5}\mu + 1} - \left( \frac{5}{3}\mu + 1 \right) \right]^{1/(2\mu)} \frac{2\pi}{a(\mu)} l_z \tag{18}$$

Following the approach in Tonttila et al. (2015), we estimate the timescale corresponding to this transition wavelength by dividing $\lambda_z$ by the collocated wind speed derived from the closest PPI scan performed by the Halo Streamline lidar.

To compare the results from this approach with what we obtain from the comparison with dissipation rates from the sonic anemometer data, we apply this technique to the data from the Halo Streamline for the whole period of XPIA, and calculate the average timescales for different stability conditions at 100m AGL. We obtain an average time scale of 32s in stable conditions, and 73s in unstable conditions. Both these values compare well with what is found with the more time-efficient comparison with the sonic anemometer retrievals (values in Table 2), thus confirming that the use of spectral models can be considered a valid alternative for the determination of the optimal sample lenghts to retrieve $\epsilon$ from lidar data.

The use of spectral models to determine the appropriate sample size to use when retrieving $\epsilon$ from lidars can also be applied when information about atmospheric stability are not available or accurate. In these cases, instead of calculating an average optimal sample size for each stability condition, an appropriate time scale can be determined at each time $\epsilon$ is retrieved from lidar measurements, from a single spectrum. We compare $\epsilon$ values from the sonic anemometers and from the Halo Streamline lidar, with the optimal time scales obtained from both the proposed approaches (comparison with the sonic anemometer data and analysis of instantaneous spectra) in Figure 10, for the same time period shown in Figure 7. The use of spectral models to determine the extension of the inertial sub-range in the lidar spectra produces valid estimates of $\epsilon$: for this case we obtain a MAE= 0.40, and a correlation coefficient $R^2 = 0.78$.

[Figure]

**Figure 9.** Example of power spectral density of the vertical component of the wind speed as measured by the Halo Streamline lidar on 11 March 2015 18:05 UTC. The red line represents the fit according to the spectral model from Eq. (16).

[Figure]

**Figure 10.** Time series from 6 April 2015 00 UTC to 10 April 2015 00 UTC comparing $\epsilon$ from sonic anemometers and the Halo Streamline lidars at 100m AGL, where the time scales for the lidars have been determined with both the proposed approaches (comparison with $\epsilon$ from sonic anemometers and fit with spectral models). Data have been smoothed with a 30-min running mean.

*References:*

- *Caughey, S.J. and Palmer, S.G., 1979. Some aspects of turbulence structure through the depth of the convective boundary layer. Quarterly Journal of the Royal Meteorological Society, 105(446), pp.811-827.*

- *Kaimal, J.C., Wyngaard, J.C.J., Izumi, Y. and Coté, O.R., 1972. Spectral characteristics of surface-layer turbulence. Quarterly Journal of the Royal Meteorological Society, 98(417), pp.563-589.*

- *Kristensen, L., Lenschow, D.H., Kirkegaard, P. and Courtney, M., 1989. The spectral velocity tensor for homogeneous boundary-layer turbulence. In Boundary Layer Studies and Applications (pp. 149-193). Springer, Dordrecht.*

- *Lothon, M., Lenschow, D.H. and Mayor, S.D., 2009. Doppler lidar measurements of vertical velocity spectra in the convective planetary boundary layer. Boundary-layer meteorology, 132(2), pp.205-226.*

- *Olesen, H.R., Larsen, S.E. and Højstrup, J., 1984. Modelling velocity spectra in the lower part of the planetary boundary layer. Boundary-Layer Meteorology, 29(3), pp.285-312.*

- *Panofsky, H.A., 1978. Matching in the convective planetary boundary layer. Journal of the Atmospheric Sciences, 35(2), pp.272-276.*

- *Tonttila, J., O'Connor, E.J., Hellsten, A., Hirsikko, A., O'Dowd, C., Järvinen, H. and Räisänen, P., 2015. Turbulent structure and scaling of the inertial subrange in a stratocumulus-topped boundary layer observed by a Doppler lidar. Atmospheric chemistry and physics, 15(10), pp.5873-5885.*

- page 2, line 5: Is the 3km scale a result from Albertson et al also? If so, move citation to end of sentence. If not, include citation for this fact also

We have eliminated the explicit reference to the 3km scale, and limited the sentence after "coarse scale". We have also added two references (Lundquist et al. 2007, Mirocha et al. 2010) at the end of the sentence.

*References:*

- *Lundquist, J.K. and Chan, S.T., 2007. Consequences of urban stability conditions for computational fluid dynamics simulations of urban dispersion. Journal of applied meteorology and climatology, 46(7), pp.1080-1097.*
- *Mirocha, J.D., Lundquist, J.K. and Kosović, B., 2010. Implementation of a nonlinear subfilter turbulence stress model for large-eddy simulation in the Advanced Research WRF model. Monthly Weather Review, 138(11), pp.4212-4228.*

- page 5, table 1: include the temporal resolution of each lidar

The temporal resolution of the lidars ($\sim$ 1 Hz) has been added to the table.

- page 7, lines 14-15: what is theta here? How small is small?

A specification about the value of theta "($< 0.1$ mrad)" has been added to the sentence.

- page 7: Are LOS velocity spectra calculated for all beams or only the vertically pointing? If all, isotropy must be assumed. Clarify and comment on this.

We have included the following sentences to clarify this point:

"For the WINDCUBE lidars, the variance of the observed line-of-sight velocity $\sigma_v^2$ can be calculated as average from all the beams. In doing so, we include turbulence contributions from both the horizontal and vertical dimensions, and we make the limiting (Kaimal et al. 1972, Mann 1994) assumption of isotropic turbulence. For the Halo Streamline lidar, which operated in a vertical stare mode, $\sigma_v^2$ is calculated from the vertically pointing beam, and therefore $\epsilon$ will strictly include turbulence contributions only in the vertical dimension, thus possibly determining different values compared to what is retrieved from the WINDCUBE lidars."

*References:*

- *Mann, J., 1994. The spatial structure of neutral atmospheric surface-layer turbulence. Journal of fluid mechanics, 273, pp.141-168.*
- *Kaimal, J.C., Wyngaard, J.C.J., Izumi, Y. and Coté, O.R., 1972. Spectral characteristics of surface-layer turbulence. Quarterly Journal of the Royal Meteorological Society, 98(417), pp.563-589.*

- page 10, lines 4-5: why is a wider inertial range expected at higher altitudes?

We have explicitly explained that this would be due to an increase in the integral scale of turbulence with height, and we have added a reference: "different altitudes can also impact the extension of the inertial sub-range, with a wider development expected at higher heights, as the integral length scale of turbulence increases (Wang et al. 2016)".
*Reference: Wang, H., Barthelmie, R.J., Doubrawa, P. and Pryor, S.C., 2016. Errors in radial velocity variance from Doppler wind lidar. Atmospheric Measurement Techniques, 9(8), p.4123.*

- page 14, line 15: this final sentence doesn't make sense. Why would the filter change the choice of shorter time scales being averaged?
We agree that the sentence can be confusing, therefore we have eliminated it.

**Technical Corrections:**

We thank the reviewer for all the suggested technical corrections, which have been incorporated in the revised version of the manuscript and supplement.

- On all figures with units of epsilon shown, use $m^2 \ s^{-3}$, not $m^2/s^3$
The plot labels have been corrected.
- When referring to figure subplots, remove space between number and letter (check AMT standard)
The space has been removed.
- Include the time scale of epsilon on all plots
The captions of figures now include the time scale of epsilon.

- Yellow lines are hard to see, especially the yellow shading. I appreciate the use of consistent colors for each instrument across all figures, but need a better choice for yellow. If v1s are used rarely, use purple or green instead of yellow, maybe the same color with different weights or dashes?
We have used green for the Halo Streamline throughout the manuscript. Yellow is not used for the WINDCUBE v1-68, which appears in a limited (2) number of figures in the main manuscript. In the supplement, yellow is now not used at all.

All the following corrections have been applied.
- line 7: accurate forecast
- 1: small enough that molecular diffusion is capable of dissipating
- 7: when using models
- 34: velocity spectra. We assess the uncertainty of this method, and present
- 5-6: as a case study... during a nocturnal low-level jet event

- 18: measurement accuracy or precision (not resolution)

- 14-15: For atmospheric stability, we classify neutral conditions as L ... unstable conditions as ... stable conditions as...

- 24: who deployed the v2? The revised sentence now includes "was deployed by the University of Colorado Boulder"

- 30: wrong dates for the v1s

- 19: remove space after tower

- 5: which must be within

- 9: define $k_1$ and N earlier

- 6: period after equation

- 22-23: different heights and atmospheric conditions

- 11: looks more like 40s, not 50s

- 24: found to be at shorter time scales than unstable

- 9-10: because they occurred less than 5% of the time

- 9: due to hard strikes

- 10: v1-61, so the comparison... 150m AGL has been performed using only this lidar's data...

- 8 conditions and smoothing

- 12: note the time scales of the raw time series We have added this specification: "(one value every ~ 4 s)".

- 20: lower case section (only capitalize when referring to Section X)

- 23: Materials, and are

- 26 & 27: space before units

- 28: intermittent

- 1: lowercase section

- 2: confusing wording

- 14: not all instruments

- 1-2: L>0 and L<0

- 4: impact on
- 11: increases of 1-2 orders (stable increases two orders)
- 1: wind energy resources
- 18: cite Yang et al, (2017). Sensitivity of turbine-height wind speeds to parameters in planetary boundary-layer and surface-layer schemes in the weather research and forecasting model. *Boundary-Layer Meteorology*, *162*(1), 117-142.

Figure 1: legend on right plot is not readable – match size of left legend. Colors on right subplot does not correspond to color scale legend on left subplot – include a new color scale for this subplot also.
The font size used in the right panel has been increased, and a new color scale has been included.
Table 1: WINDCUBE v1 (61 & 68)
Corrected.
Figure 5: Use a different color than yellow (purple?)
Purple is now used.
Figure 6: variability (misspelled); indicate which time scale is used for each stability class: minimum MAE for optimized Nt, at the appropriate time scales?
The caption has been modified accordingly.
Figure 8: labels columns (raw and smoothed) and rows (all stability, stable, unstable) on figure
Labels have been added.
Figure 9: label instruments on figures; nighttime variability mentioned in text is hard to see on this color scale – maybe change to "jet" blue-red scale; I'd prefer the y-axes to be the same on the two left plots, or at least both start at 0; are these 30-minute or raw values?
We have labeled the instruments on the plots. Thank you for your suggestion about the color scale. However, we have decided to keep the current color scale, as 'jet' can create some confusion, especially when printed in black and white, as shown in Light and Bartlein 2004 and Stauffer et al. 2015.
We have now used a common vertical axis for all the panels as suggested.
Raw values are used, and this is now specified in the caption of the Figure ("Daily climatology of turbulence dissipation rate derived from raw values …").
*References:*

- *Light, A. and Bartlein, P.J., 2004. The end of the rainbow? Color schemes for improved data graphics. Eos, Transactions American Geophysical Union, 85(40), pp.385-391.*
- *Stauffer, R., Mayr, G.J., Dabernig, M. and Zeileis, A., 2015. Somewhere over the rainbow: How to make effective use of colors in meteorological visualizations. Bulletin of the American Meteorological Society, 96(2), pp.203-216.*

**RESPONSE TO REVIEWER #2**

*In this document, the reviewer comments are in black, the authors responses are in red.*

The authors thank the reviewer for their detailed review and useful suggestions to improve the quality of our work.

In this manuscript, a technique to measure turbulence dissipation rate from Doppler lidar observations is presented using data collected from several Doppler lidars during XPIA. The dissipation rates are compared with those from sonic anemometers for verification (and to determine the sample length for the best agreement). Statistics of dissipation are presented for the experiment, which serve as a brief climatology of dissipation at the site.

The manuscript is generally written and organized well, and results in this manuscript are of significant interest to a wide audience in the Doppler lidar and boundary-layer fields. Still, there are some significant omissions in the description of the technique and how the presented results are interpreted. As such, I recommend that this manuscript be reconsidered for publication after major revisions, after the following concerns have been addressed.

Thank you for finding our results interesting!

**General/major comments:**

a) How exactly is the turbulence dissipation calculated using the Doppler lidar data? More details need to be added to Sect. 3.2 so that this technique could be applied by a reader. From the Halo data, it must be the vertical staring observations. From the V1/V2 profiler data, which beam position is used (and why)? Was dissipation calculated from each beam separately, and the mean of those used?

While isotropy is assumed, turbulence is rarely isotropic in the boundary-layer, especially under stable conditions when turbulent eddies are more horizontally oriented. As such, there could be differences (particularly with the Halo which just uses vertical beam) between the lidar estimates and sonic anemometer estimates (which use the horizontal variance alone) from anisotropy. This should be briefly discussed.

The description of the method in Section 3.2 now includes the following sentences, which also briefly comment the assumption of isotropic turbulence:

"For the WINDCUBE lidars, the variance of the observed line-of-sight velocity $\sigma_v^2$ can be calculated as average from all the beams. In doing so, we include turbulence contributions from both the horizontal and vertical dimensions, and we make the limiting (Kaimal et al. 1972, Mann 1994) assumption of isotropic turbulence. For the Halo Streamline lidar, which operated in a vertical stare mode, $\sigma_v^2$ is calculated from the vertically pointing beam, and therefore $\epsilon$ will strictly include turbulence contributions only in the vertical dimension,

thus possibly determining different values compared to what is retrieved from the WINDCUBE lidars. Another difference due to the different scan patterns used by the considered lidars is related to the determination of the horizontal wind speed U. For the WINDCUBE lidars, U can be derived from the line-of-sight velocity measurements from the different beams, with the assumption of horizontal homogeneity of the flow over the probed volume. In the case of the Halo Streamline, no information about the horizontal wind can be derived from the measurements in the vertical staring mode, which only measures the vertical component of the wind speed. U is then retrieved from a sine-wave fitting from the VAD scans that are performed every 12 min".

Moreover, we have added and modified the following sentences in Section 4, to emphasize again the differences between the results from the different instruments:

"It is reasonable to explain the higher error ($\sim$ +10%) of the Halo Streamline compared to the WINDCUBE lidars at 100m AGL as a consequence of the differences in the spatial dimensions that are samples by the two lidars. While the lidar beams of the WINDCUBE are tilted, and they therefore include turbulence contributions in the horizontal dimension (which is the only contribution considered in the determination of $\epsilon$ from the sonic anemometers), $\epsilon$ from the Halo Streamline is only retrieved using information from the vertically pointing beams. Moreover, the necessary approximations adopted in the determination of the horizontal velocity $U$ for the Halo Streamline lidar, as explained in Section 3.2, likely determine an additional error increase for this lidar."

*References:*

- *Mann, J., 1994. The spatial structure of neutral atmospheric surface-layer turbulence. Journal of fluid mechanics, 273, pp.141-168.*
- *Kaimal, J.C., Wyngaard, J.C.J., Izumi, Y. and Coté, O.R., 1972. Spectral characteristics of surface-layer turbulence. Quarterly Journal of the Royal Meteorological Society, 98(417), pp.563-589.*

b) In Sect. 4, the sampling length for calculation of dissipation during stable, neutral, and unstable conditions is chosen as the minimum of the MAE between the sonic and lidar estimate. This is fine when there is sonic anemometer data for both verification and classifying stability, but most sites that this technique could be applied to will not have coincident sonic measurements. How could this technique be applied to other sites, where the turbulence characteristics/stability might be quite different? This is a major limiting factor in the applicability of this technique, and currently there is no discussion of how this could be applied to other sites given this limitation. Also, does the minimum in the MAE vary between slightly stable and strongly stable conditions, when the inertial subrange may be much smaller? Should the analysis in Fig. 5 be done with more stability classifications (strongly stable/unstable, weakly stable/unstable, neutral)? Perhaps this technique could be refined so that the sample length varies with the outer scale of the inertial subrange, as determined from the Doppler lidar data alone. Then, the technique could be easily applied

to other lidar data. Alternatively, the authors could add a short section (a few paragraphs) on how this technique could be applied at locations without sonic anemometer data for stability and determination of the sample length to use.

We have refined our approach to propose an alternative to use when measurements from co-located sonic anemometers are not available. We have included in the manuscript the following additional subsection:

**4.1 Determination of the optimal time scales to retrieve $\epsilon$ from lidars in absence of co-located sonic anemometers**

The availability of multiple sonic anemometers co-located with the lidars at XPIA has allowed for a direct comparison between $\epsilon$ estimates from different instruments to determine the optimal length scales, in different stability conditions, to use when retrieving $\epsilon$ from Doppler lidar measurements. This approach does not require the direct calculation of spectra from the line-of-sight velocity measured by the lidars, and therefore it represents a time-efficient technique. However, the proposed method is only viable when sonic anemometers are deployed in the near vicinity of a lidar, and when measures of atmospheric stability are available.

When a comparison with sonic anemometer data is not possible, the appropriate time scale to use in the lidar retrieval of $\epsilon$ can be determined by finding the maximum wavelength within the inertial sub-range in the velocity spectra from the lidar measurements. To do so, spectral models can be fitted to the observed spectra. Several models have been proposed for turbulence spectra in different stability conditions (Kaimal et al., 1972; Panofsky, 1978; Olesen et al., 1984). We test the spectral model proposed by Kristensen et al. (1989), which proposes expressions for both the cases of an isotropic and an anisotropic horizontally homogeneous flow. To validate our results and test this alternative approach to derive $\epsilon$ from lidar measurements, we use data from the Halo Streamline lidar to estimate the maximum wavelength $\lambda_z$ within the inertial subrange. Since the Halo mainly operated in a vertical stare mode during XPIA, we consider the following expression for the turbulence spectrum of the vertical component of the wind speed:

$$S(k) = \frac{\sigma_z^2 l_z}{2\pi} \frac{1 + \frac{8}{3}\left(\frac{l_z k}{a(\mu)}\right)^{2\mu}}{\left[1 + \left(\frac{l_z k}{a(\mu)}\right)^{2\mu}\right]^{5/(6\mu)+1}} \tag{16}$$

where $k$ is the wavenumber, $\sigma_z$ is the standard deviation of the vertical component of the wind speed used to compute the spectrum, $l_z$ is the integral scale of the vertical velocity along the horizontal flow trajectory, and the parameter $\mu$ controls the curvature of the spectrum. We use $\mu = 1.5$, which provides a good match with our experimental spectra, as also found in previous studies (Lothon et al., 2009; Tonttila et al., 2015). The parameter $a$ can be expressed as a function of $\mu$ as:

$$a(\mu) = \pi \frac{\mu\Gamma\left(\frac{5}{6\mu}\right)}{\Gamma\left(\frac{1}{2\mu}\right)\Gamma\left(\frac{1}{3\mu}\right)} \tag{17}$$

We calculate spectra using 10-min consecutive data, and we fit the spectral model to the experimental data, leaving out frequencies greater than 0.2Hz, which are affected by instrumental noise (Frehlich, 2001), not modeled here. An example of a measured spectrum and the fit resulting from the model are shown in Figure 9. The transition wavelength $\lambda_z$ between the

[Figure]

**Figure 9.** Example of power spectral density of the vertical component of the wind speed as measured by the Halo Streamline lidar on 11 March 2015 18:05 UTC. The red line represents the fit according to the spectral model from Eq. (16).

inertial sub-range and the outer scales can be expressed as a function of the integral scale $l_z$ and the parameter $\mu$:

$$\lambda_z = \left[ \frac{5}{3}\sqrt{\mu^2 + \frac{6}{5}\mu + 1} - \left(\frac{5}{3}\mu + 1\right) \right]^{1/(2\mu)} \frac{2\pi}{a(\mu)} l_z \tag{18}$$

Following the approach in Tonttila et al. (2015), we estimate the timescale corresponding to this transition wavelength by dividing $\lambda_z$ by the collocated wind speed derived from the closest PPI scan performed by the Halo Streamline lidar.

To compare the results from this approach with what we obtain from the comparison with dissipation rates from the sonic anemometer data, we apply this technique to the data from the Halo Streamline for the whole period of XPIA, and calculate the average timescales for different stability conditions at 100m AGL. We obtain an average time scale of 32s in stable conditions, and 73s in unstable conditions. Both these values compare well with what is found with the more time-efficient comparison with the sonic anemometer retrievals (values in Table 2), thus confirming that the use of spectral models can be considered a valid alternative for the determination of the optimal sample lenghts to retrieve $\epsilon$ from lidar data.

The use of spectral models to determine the appropriate sample size to use when retrieving $\epsilon$ from lidars can also be applied when information about atmospheric stability are not available or accurate. In these cases, instead of calculating an average optimal sample size for each stability condition, an appropriate time scale can be determined at each time $\epsilon$ is retrieved from lidar measurements, from a single spectrum. We compare $\epsilon$ values from the sonic anemometers and from the Halo Streamline lidar, with the optimal time scales obtained from both the proposed approaches (comparison with the sonic anemometer data and analysis of instantaneous spectra) in Figure 10, for the same time period shown in Figure 7. The use of spectral models to determine the extension of the inertial sub-range in the lidar spectra produces valid estimates of $\epsilon$: for this case we obtain a MAE= 0.40, and a correlation coefficient $R^2 = 0.78$.

[Figure]

**Figure 10.** Time series from 6 April 2015 00 UTC to 10 April 2015 00 UTC comparing $\epsilon$ from sonic anemometers and the Halo Streamline lidars at 100m AGL, where the time scales for the lidars have been determined with both the proposed approaches (comparison with $\epsilon$ from sonic anemometers and fit with spectral models). Data have been smoothed with a 30-min running mean.

**Specific comments:**

a) p. 2 line 5; p. 21 line 27: Here, the authors make the case that both production and dissipation of TKE need to be known for turbulence closure. The authors state that by measuring dissipation, the scales at which the assumption of local equilibrium are broken will be assessed. However, in order to do this, production must also be measured. The authors should add a few statements on how production of TKE can be measured for the full closure.

We agree that TKE production needs to be calculated in order to have a full closure of the TKE budget. Since the focus of this work is on determining the variability of turbulence dissipation, which itself has an extreme importance as shown in Yang et al. 2017, we have decided to leave out from this manuscript the reference to the determination of the scales at which the assumption of local equilibrium breaks. As a consequence, we have deleted from the introduction the sentence "in order to understand at what spatio-temporal scale local imbalance becomes important." We have also deleted from the conclusions the sentence "the scales at which the assumption of local equilibrium is broken will be assessed".

b) Figure 1 caption: Would be good to clarify that contours in the right panel are in m.

The caption of the figure now includes: "Contours in the right panel show elevation in m ASL."

c) p 3 line 1: Spell out XPIA in full here, for those unfamiliar with the project.

We have included "eXperimental Planetary boundary layer Instrumentation Assessment" in the revised version.

d) p. 4, line 2: Was this sonic also a CSAT3 or was it different? If it was a different type of sonic, are there differences in the design that may cause the observed dissipation to be much higher than for CSAT 3 (possibly more obstructions, if it's an RM Young anemometer) as later discussed in Sect. 5? Given its importance to the results, more details should be provided about this sonic, its siting, and any QC applied to it (was any data thrown out when it was waked by what it was mounted on)?

The sonic at 5m AGL was a CSAT3 as well. The description of this instrument in Section 2.1 is now as follows: "An additional sonic anemometer was mounted on a 5-m AGL surface flux station located 200 m south-west of the BAO tower over natural arid grassland. The sonic anemometer (Campbell CSAT3A) at this location operated with a frequency of 10 Hz." The location of this 5m sonic anemometer is now included in the map in Figure 1.

e) Table 1: Can the pulse width (FWHM) be added as a row to this table, as well? This will be useful in understanding the smallest eddies that can be resolved by a given lidar.

The Table now includes the pulse width for the instruments: 200ns for the WINDCUBE v1s, 175ns for the WINDCUBE v2, 150ns for the Halo Streamline.

f) p. 5 line 19: How did the measured dissipation rates between the two sonic anemometers compare to each other when both were unwaked? Were they often similar, or were there often substantial differences? This might be useful to form a 'baseline' estimate of how much uncertainty is in any dissipation measurement from the sonic anemometers themselves.

We have compared dissipation rates from the two sonics (at each of the 6 heights of the BAO tower), and added the following sentences at the end of Section 3.1:

"As already mentioned, data were excluded for wind directions waked by the tower. When neither of the two anemometers is affected by tower wakes, $\epsilon$ is defined as the average between the two independent values obtained from the two sonics at each height. To quantify the uncertainty in turbulence dissipation rate measurements from the sonic anemometers, we have compared $\epsilon$ from the two sonics at each level when neither one was influenced by the tower wake. For each tower boom direction (northwest and southeast), we calculate the median absolute error (MAE) between $\epsilon$ from the sonic anemometers mounted on the considered boom direction and the correspondent average value from the two sonics:

$$MAE = median\left(\frac{|\epsilon_{single} - \epsilon_{average}|}{\epsilon_{average}}\right)$$

In calculating the error, we consider data from all heights, as no significant difference was noticed at different levels. For both the boom directions, we find very similar results, with MAE = 0.19, which is reduced to 0.14 when a 30-min running mean is applied to the $\epsilon$ time series. The distributions of the errors are included in the Supplementary Material. No bias was detected between the retrievals from the sonic anemometers on the two boom directions."

We have also included the following plots in the Supplementary Material:

[Figure]

Figure S2: (a) histogram of the fractional median error between turbulence dissipation rate calculated from the sonic anemometers on the northwest booms and the average

dissipation from both the boom directions. Results for the sonic anemometers on the southeast booms are similar. (b) as in (a), but median absolute error. Raw values of $\epsilon$ are used.

g) Eq. 5: By using this equation to estimate dissipation, it is implicitly assumed that the line-of-sight atmospheric variance ($\sigma w2$ in Eq. 8) is strictly the result of turbulent motion. However, non-turbulent motions such as gravity waves in a stable layer may increase the line-of-sight variance but are not turbulent, thus there is little dissipation with them. Under these conditions, turbulence dissipation would be overstated. This may be especially important at the BAO for westerly winds, due to the close presence of mountains to the west that may induce mountain waves when the atmospheric conditions permit. This may affect the statistics later presented in Sect. 5, as dissipation may be overestimated due to the presence of these waves.

Given the extremely short time scales we are considering in our calculations (usually < 2min), we think that it is a reasonable assumption to assimilate, at the considered time scales, the increase in variance due to gravity waves to turbulent motions. Such a contamination of the strictly turbulent component of the motion from larger processes is somehow unavoidable and implicitly assumed in a variety of boundary layer calculations, for example when picking the averaging time scale to calculate Reynolds decompositions. Moreover, even when calculating turbulence dissipation rates with the traditional spectral technique from sonic anemometers, the same contamination would take place.

In the manuscript, we have made this assumption explicit as follows: "By assuming that the contribution of all atmospheric flows to the observed line-of-sight variance within the considered short time scales can be regarded as of turbulent nature, the variance $\sigma_v^2$ in (7) can be written as the sum of three different terms".

h) Eq. 8: In the term $\sigma w2$ it should be clarified that this is not the true atmospheric variation of the wind, as the smallest scales of turbulence are not resolved by the lidar.

We have modified the sentence as "$\sigma_w^2$ is the desired net contribution from atmospheric turbulence at scales that can be measured by the lidar (Brugger et al. 2016)."

*Reference: Brugger, P., Träumner, K. and Jung, C., 2016. Evaluation of a procedure to correct spatial averaging in turbulence statistics from a Doppler lidar by comparing time series with an ultrasonic anemometer. Journal of Atmospheric and Oceanic Technology, 33(10), pp.2135-2144.*

i) p. 8 line 23: Do the line-of-sight velocities need to be de-trended? Since the windows over which the variance is calculated is short (<1 min), the de-trending will effectively remove variance contributions from large eddies, especially during unstable conditions, causing an underestimate of variance (and consequently dissipation).

When using not-detrended data, the minimum error in the $\epsilon$ comparison lidar – sonics is about 10% higher than what we got with the de-trended data. Therefore, we decided to stick with the traditional (in statistics) approach of detrending time series before applying spectral analysis.

j) p. 9 line 5: Since the sampling window is so short, measurement uncertainty/ representativeness (i.e., Lenschow et al 1994) is a significant factor in the quality/error of the variance measurement as well and should be mentioned.

We have modified the sentence as follows: "In fact, the shorter the sampling time, the higher the measurement error in the estimate of the variance of line-of-sight velocity would be, because of both higher measurement uncertainty which impacts its representativeness (Lenschow et al. 1994) and a higher relative contribution of the instrumental noise."

k) Figure 4: Could vertical lines be added to denote the inertial subrange and/or sample length used?

We have included the following sentence in the caption of the Figure: "To calculate $\epsilon$ for these cases, the optimal sample length from comparison with the sonic anemometers corresponds to frequencies greater than $0.04 s^{-1}$ for stable conditions, greater than $0.01 s^{-1}$ for unstable conditions."

l) p. 10 line 15: Could an equation be included here for how exactly the metric presented in Fig. 5 is calculated? I assume the error is normalized by some value (as the y-axis is unitless), but this is unclear. Without this information, it is difficult to interpret Fig. 5. The caption for Fig. 5 needs to be clarified accordingly, as well.

We have added a sentence to define the metric used: "To quantify the difference between sonic and lidar estimates of $\epsilon$, we use the median absolute error (MAE), defined as:

$$MAE = median\left(\frac{|\epsilon_{lidar} - \epsilon_{sonic}|}{\epsilon_{sonic}}\right)$$

"

We have also modified the y-axis label of the plot as "Fractional median absolute error".

m) p. 14 line 33: As SNR typically decreases with range, is it possible that the increase in dissipation above 600 m is due to more noisy/random errors in the line-of-sight measurements above 600 m? Thus, the increase above this height is not real (due to atmospheric turbulence), but instead due to increasing measurement errors.

A SNR threshold has been set to QC the data, however we agree that the average SNR aloft is lower, even after setting a threshold. Our data also show that we mostly had valid data aloft during high wind conditions. Therefore, we have modified the sentence as: "The slight increase of $\epsilon$ above $\sim$ 600m AGL at night for the Halo Streamline lidar can be explained as due to more random errors in the line-of-sight velocity measured by the lidar at high

altitudes but also as effect of the higher frequency of good-quality measurements at higher levels during high wind speed events".

n) Figure 9: The labels on these plots are small and difficult to read. Could they be made larger?
The labels are now bigger.

o) p. 17 line 1: Are there other studies that also confirm the finding here that there is a significant gradient in dissipation right near the ground, but the changes are much smaller above 50 m? What physically results in this large almost order of magnitude change in dissipation from the surface upwards? It would be good to expand on this. Without justification or other studies that show similar results, these results seem a little suspect. Was the 5-m sonic near anything that may obstruct the flow to cause dissipation to be so large?
No obstacle was located near the 5-m sonic. We have added a reference to the sentence to show that our results are consistent with what was found in previous studies: "The plot confirms that turbulence dissipation rate shows most of its variability with height close to the surface, as also found by Balsley et al. 2006."
We expect the increase in dissipation close to the surface to be connected with the increased TKE shear production close to the surface. Therefore, we have added the following sentence: "We expect this large reduction in $\epsilon$ to be due to a rapid decrease in shear production with height close to the surface, as it has been shown (Nilsson et al. 2016) that shear production has a strong connection with dissipation close to the surface."

[revised manuscript text omitted]
. To quantify the uncertainty in turbulence dissipation rate measurements from the sonic anemometers, we have compared $\epsilon$ from the two sonics at each level when neither one was influenced by the tower wake. For each tower boom direction (northwest and southeast), we calculate the median absolute error (MAE) between $\epsilon$ from the sonic anemometers

[Figure]

**Figure 2.** Turbulence energy spectrum according to Kolmogorov's hypothesis.

mounted on the considered boom direction and the correspondent average value from the two sonics:

$$\text{MAE} = \text{median} \left( \frac{|\epsilon_{single} - \epsilon_{average}|}{\epsilon_{average}} \right) \tag{4}$$

In calculating the error, we consider data from all heights, as no significant difference was noticed at different levels. For both the boom directions, we find very similar results, with $\text{MAE} = 0.19$, which is reduced to $0.14$ when a 30-min running mean is

5    applied to the $\epsilon$ time series. The distributions of the errors are included in the Supplementary Material. No bias was detected between the retrievals from the sonic anemometers on the two boom directions.

**3.2   Dissipation from Doppler lidar**

Wind Doppler lidars can provide a great improvement of our understanding of the variability of turbulence dissipation thanks to the ease of their deployment in different locations and the long measurement range allowed by several commercial models.

10   To do so, robust methods to estimate $\epsilon$ with lidars are necessary, and their uncertainty has to be assessed. For this purpose, we follow and refine the  approach described in O'Connor et al. (2010) to estimate $\epsilon$ from vertical profiling lidars or scanning lidars used in vertical staring mode. For homogeneous and isotropic turbulence, within the inertial sub-range, the turbulent energy spectrum (Figure 2) can be expressed according to the Kolmogorov (1941) hypothesis in terms of wavenumber $k$ as:

$$S(k) = a\epsilon^{2/3}k^{-5/3} \tag{5}$$

15   where $a \simeq 0.52$ is the one-dimensional Kolmogorov constant. The wavenumber $k$ can be written in terms of a length scale $L = 2\pi/k$ by invoking Taylor's frozen turbulence hypothesis (Taylor, 1935). By integrating (5) over the wavenumber space,  starting from the wavenumber $k_1$ correspondent to a single lidar sample, the variance $\sigma_v^2$ of the de-trended observed lineof-sight velocity from $N$ samples can be obtained:

$$\sigma_v^2 = \int\limits_k^{k_1} S(k)dk = -\frac{3}{2}a\epsilon^{2/3}\left(k_1^{-2/3} - k^{-2/3}\right) = \tag{6}$$

$$= \frac{3a}{2}\left(\frac{\epsilon}{2\pi}\right)^{2/3}\left(L_N^{2/3} - L_1^{2/3}\right) \tag{7}$$

and therefore if the length scales are properly chosen (and consistent with how $\sigma_v$ is computed) then $\epsilon$ can be calculated without the need of systematically computing turbulence energy spectra. In (7), the length scale $L_1$ for a single sample interval is given by:

$$L_1 = Ut + 2z\sin\left(\frac{\theta}{2}\right) \tag{8}$$

where $U$ is the horizontal wind speed, $t$ is the dwell time, $\theta$ the half-angle divergence of the lidar beam, and $z$ the height AGL. Since Doppler lidars generally have a very small $\theta$ ($< 0.1\,\mathrm{mrad}$), the second term in (8) is typically negligible. For $N$ samples, the length scale becomes $L_N = NUt$.  For the WINDCUBE lidars, the variance of the observed line-of-sight velocity $\sigma_u^2$ can be calculated as average from all the beams. In doing so, we include turbulence contributions from both the horizontal and vertical dimensions, and we make the limiting (Kaimal et al., 1972; Mann, 1994) assumption of isotropic turbulence. For the Halo Streamline lidar, which operated in a vertical stare mode, $\sigma_u^2$ is calculated from the vertically pointing beam, and therefore $\epsilon$ will strictly include turbulence contributions only in the vertical dimension, thus possibly determining different values compared to what is retrieved from the WINDCUBE lidars. Another difference due to the different scan patterns used by the considered lidars is related to the determination of the horizontal wind speed $U$. For the WINDCUBE lidars, $U$ can be derived from the line-of-sight velocity measurements  from the different beams, with the assumption of horizontal homogeneity of the flow over the probed volume. In the case of the Halo Streamline, no information about the horizontal wind can be derived from the measurements in the vertical staring mode, which only measures the vertical component of the wind speed. $U$ is then retrieved from a sine-wave fitting from the VAD scans that are performed every 12min. The heights at which the measurements are taken during the tilted VAD scans are not the same as the heights sampled in the vertical staring mode. Therefore, for each considered level in the vertical staring mode, $U$ is determined from a linear interpolation of the wind speed retrieved at the two closest heights during the VAD scans. Considerations about the error introduced by this procedure on the estimation of $\epsilon$ will be discussed in Section 4.

Lidar measurements are inherently affected by signal noise as well as possible variations of the aerosol fall speeds, which provide additional contributions to the observed variance.  By assuming that the contribution of all atmospheric flows to the observed line-of-sight variance within the considered short time scales can be regarded as of turbulent nature, the variance $\sigma_v^2$ in (7) can be written as the sum of three different terms, which can be considered to be independent of one other (Doviak et al., 1993):

$$\sigma_v^2 = \sigma_w^2 + \sigma_e^2 + \sigma_d^2 \tag{9}$$

[revised manuscript text omitted]

**Figure 4.** Turbulence energy spectrum for a stable case (panel a - 2 April 2015, 03:00 UTC), and an unstable case (panel b - 3 April 2015, 22:15 UTC), calculated from 15 minutes of data measured by the WINDCUBE v2 at 100m AGL. The dashed lines represent the theoretical $-5/3$ slope expected in the inertial sub-range. To calculate $\epsilon$ for these cases, the optimal sample length from comparison with the sonic anemometers corresponds to frequencies greater than $0.04\text{s}^{-1}$ for stable conditions, greater than $0.01\text{s}^{-1}$ for unstable conditions.

atmospheric stability. Figure 4 shows examples of turbulence spectra calculated over 15-min intervals for data measured by the WINDCUBE v2 lidar at $100$m AGL in different stability conditions. For stable conditions (panel a), the transition from the inertial sub-range (which can be identified by comparing the slope of the spectrum with the theoretical $-5/3$ value shown by the dashed line) to the outer scales occurs at a higher frequency compared to the unstable case (panel b). Therefore, the choice
5 of the number of samples $N$ to use in the calculation should change accordingly. As a general rule, we expect shorter time scales to be adequate for stable conditions, when the turbulent eddies in the boundary layer are smaller, while longer scales would be more suitable during unstable conditions, characterized by larger convective eddies that can be fully captured only when using larger scales. Moreover, different altitudes can also impact the extension of the inertial sub-range, with a wider development expected at higher heights, as the integral length scale of turbulence increases (Wang et al., 2016).
10 To estimate the appropriate time scales which best balance these competing factors, we calculate $\epsilon$, at each height from each of the considered lidars, using several values for the number of samples $N$ used in the calculation. At the heights where there is correspondence between lidar and sonic anemometer measurements, we then compare the $\epsilon$ values from the lidars with the corresponding $\epsilon$ calculated at the meteorological tower. The estimates of $\epsilon$ from sonic anemometers and lidars have been calculated at slightly different time stamps, given the unavoidable difference in the nominal measurement time stamps of instruments operating with different temporal resolutions. Given the inherent turbulent nature of $\epsilon$ and its remarkable range
15 of variability, the comparison between the time series from sonic anemometers and lidars could be flawed by the effect of the turbulent high-frequency variability of $\epsilon$. Moreover, since this analysis is focused on the assessment of the appropriate time scales for different stability conditions, consistency with the time scale used to calculate turbulent fluxes for the determination of the Obukhov Length $L$ is advisable. Therefore, a 30-min running mean is applied to the time series of $\epsilon$ from both sonic
20 anemometers and lidars before comparing the estimates from the different instruments.

[Figure]

**Figure 5.** Median absolute error between $\epsilon$ estimates (smoothed with a 30-min running mean) from sonic anemometer and WINDCUBE v2 lidar data at $100$m AGL during the whole period of the XPIA campaign, as a function of the sample length used to estimate $\epsilon$ from lidar data.

To quantify the difference between sonic and lidar estimates of $\epsilon$, we use the median absolute error (MAE), defined as:

$$\text{MAE} = \text{median}\left(\frac{|\epsilon_{lidar} - \epsilon_{sonic}|}{\epsilon_{sonic}}\right) \tag{15}$$

[revised manuscript text omitted]

are not reported. For the WINDCUBE lidars, the median absolute error slightly increases with height, likely because of the severe reduction of the number of acceptable measurements at higher levels, and it always stays below 50%. For the Halo Streamline lidar, the median error stays almost constant in the considered portion of the boundary layer. It is reasonable to explain the higher error ($\sim +10\%$) of the Halo Streamline compared to the  WINDCUBE lidars at 100m AGL as

5  a consequence of the differences in the spatial dimensions that are samples by the two lidars. While the lidar beams of the WINDCUBE are tilted, and they therefore include turbulence contributions in the horizontal dimension (which is the only contribution considered in the determination of $\epsilon$ from the sonic anemometers), $\epsilon$ from the Halo Streamline is only retrieved using information from the vertically pointing beams. Moreover, the necessary approximations adopted in the determination of the horizontal velocity $U$ for the Halo Streamline lidar, as explained in Section 3.2, likely determine an additional error

10  increase for this lidar. However, the magnitude of this additional error due to the reduced frequency in determining $U$ for the Halo Streamline is comparable with the additional uncertainty related to the drop of instrumental performance that the WINDCUBE show at higher levels. Therefore, the estimates of $\epsilon$ from the Halo Streamline can be considered physically robust in the lowest few hundred meters of the boundary layer.

Possible sources for the discrepancy found between $\epsilon$ from sonic anemometers and lidars might arise from the different

15  temporal resolution and sampling volumes of the various instruments, as well as the 100m spatial separation between the lidar site and the BAO meteorological tower. In any case, given the wide range of variability of $\epsilon$, which can span $\sim 6$ orders of magnitude during its typical diurnal cycle (Section 5),  and the inherent uncertainty in $\epsilon$ retrievals even from just the sonic anemometers (Section 3.1), the obtained magnitudes of the error prove that the refined method to retrieve $\epsilon$ from lidar measurements gives robust estimates of turbulence dissipation rate. The accommodation for different stability conditions in the

[Figure]

**Figure 7.** Time series from 6 April 2015 00 UTC to 10 April 2015 00 UTC comparing $\epsilon$ from sonic anemometers and all the considered lidars at 100m AGL. Data have been smoothed with a 30-min running mean.

choice of the time scales used in the method considerably reduces, especially for stable conditions, the magnitude of the errors (obtained through propagation of errors) found in the original study (O'Connor et al., 2010). To visualize the good agreement between sonic anemometer and lidar estimates of $\epsilon$, Figure 7 shows the time series for a portion of the XPIA campaign, with values from all the considered instruments at 100m AGL. A clear diurnal pattern is revealed, with higher values of turbulence dissipation during the day, and differences of several orders of magnitude between daytime and nighttime values of $\epsilon$. These results will be explored in more detail in Section 5. A systematic comparison between $\epsilon$ estimates from sonic anemometers and the WINDCUBE v2 lidar at 100m AGL is shown by the density histograms in Figure 8, for the whole period of the XPIA campaign, for different stability conditions and smoothing. The coefficient of determinations $R^2$ are also reported in the plots. The good agreement between data from sonic anemometer and lidars is confirmed, with unstable conditions showing a better performance ($R^2 = 0.89$ for the smoothed time series) compared to stable conditions ($R^2 = 0.74$). Moreover, the plots show the effect of the choice of applying the 30-min running mean before comparing $\epsilon$ values from the different instruments. In the figure, the panels on the left compare $\epsilon$ without any temporal filter (one value every $\sim 4$s), while the panels on the right show the comparison between time series after the 30-min running mean has been applied. The application of the 30-min running mean to the $\epsilon$ time series increases the correlation between the different time series. In any case, even for the raw time series, the values of the coefficient of determination are always greater than $0.6$.

Also, the application of this filter does not considerably change the choice of the appropriate time scales.

**4.1 Determination of the optimal time scales to retrieve $\epsilon$ from lidars in absence of co-located sonic anemometers**

RAW                                    SMOOTHED

[Figure]

**Figure 8.** Correlation between $\epsilon$ values from sonic anemometer and WINDCUBE v2 lidar at 100m AGL for the whole period of the XPIA campaign, using the selected time scales for the estimation of $\epsilon$ from lidar data. The color scales represent the probability of occurrence in percentage, and the dark dashed lines show perfect correlation. (a) All stability conditions, raw data (MAE = 62%); (b) all stability conditions, 30-min running mean applied (MAE = 34%); (c) stable conditions, raw data (MAE = 67%); (d) stable conditions, 30-min running mean applied (MAE = 44%); (e) unstable conditions, raw data (MAE = 58%); (f) unstable conditions, 30-min running mean applied (MAE = 27%).

The availability of multiple sonic anemometers co-located with the lidars at XPIA has allowed for a direct comparison between $\epsilon$ estimates from different instruments to determine the optimal length scales, in different stability conditions, to use when retrieving $\epsilon$ from Doppler lidar measurements. This approach does not require the direct calculation of spectra from the line-of-sight velocity measured by the lidars, and therefore it represents a time-efficient technique. However, the proposed

method is only viable when sonic anemometers are deployed in the near vicinity of a lidar, and when measures of atmospheric stability are available.

When a comparison with sonic anemometer data is not possible, the appropriate time scale to use in the lidar retrieval of $\epsilon$ can be determined by finding the maximum wavelength within the inertial sub-range in the velocity spectra from the lidar measurements. To do so, spectral models can be fitted to the observed spectra. Several models have been proposed for turbulence spectra in different stability conditions (Kaimal et al., 1972; Panofsky, 1978; Olesen et al., 1984). We test the spectral model proposed by Kristensen et al. (1989), which proposes expressions for both the cases of an isotropic and an anisotropic horizontally homogeneous flow. To validate our results and test this alternative approach to derive $\epsilon$ from lidar measurements, we use data from the Halo Streamline lidar to estimate the maximum wavelength $\lambda_z$ within the inertial subrange. Since the Halo mainly operated in a vertical stare mode during XPIA, we consider the following expression for the turbulence spectrum of the vertical component of the wind speed:

$$S(k) = \frac{\sigma_z^2 l_z}{2\pi} \frac{1 + \frac{8}{3}\left(\frac{l_z k}{a(\mu)}\right)^{2\mu}}{\left[1 + \left(\frac{l_z k}{a(\mu)}\right)^{2\mu}\right]^{5/(6\mu)+1}} \tag{16}$$

where $k$ is the wavenumber, $\sigma_z$ is the standard deviation of the vertical component of the wind speed used to compute the spectrum, $l_z$ is the integral scale of the vertical velocity along the horizontal flow trajectory, and the parameter $\mu$ controls the curvature of the spectrum. We use $\mu = 1.5$, which provides a good match with our experimental spectra, as also found in previous studies (Lothon et al., 2009; Tonttila et al., 2015). The parameter $a$ can be expressed as a function of $\mu$ as:

$$a(\mu) = \pi \frac{\mu \Gamma\left(\frac{5}{6\mu}\right)}{\Gamma\left(\frac{1}{2\mu}\right)\Gamma\left(\frac{1}{3\mu}\right)} \tag{17}$$

We calculate spectra using 10-min consecutive data, and we fit the spectral model to the experimental data, leaving out frequencies greater than 0.2Hz, which are affected by instrumental noise (Frehlich, 2001), not modeled here. An example of a measured spectrum and the fit resulting from the model are shown in Figure 9. The transition wavelength $\lambda_z$ between the inertial sub-range and the outer scales can be expressed as a function of the integral scale $l_z$ and the parameter $\mu$:

$$\lambda_z = \left[\frac{5}{3}\sqrt{\mu^2 + \frac{6}{5}\mu + 1} - \left(\frac{5}{3}\mu + 1\right)\right]^{1/(2\mu)} \frac{2\pi}{a(\mu)} l_z \tag{18}$$

Following the approach in Tonttila et al. (2015), we estimate the timescale corresponding to this transition wavelength by dividing $\lambda_z$ by the collocated wind speed derived from the closest PPI scan performed by the Halo Streamline lidar.

To compare the results from this approach with what we obtain from the comparison with dissipation rates from the sonic anemometer data, we apply this technique to the data from the Halo Streamline for the whole period of XPIA, and calculate the average timescales for different stability conditions at 100m AGL. We obtain an average time scale of 32s in stable conditions,

[Figure]

**Figure 9.** Example of power spectral density of the vertical component of the wind speed as measured by the Halo Streamline lidar on 11 March 2015 18:05 UTC. The red line represents the fit according to the spectral model from Eq. (16).

and 73s in unstable conditions. Both these values compare well with what is found with the more time-efficient comparison with the sonic anemometer retrievals (values in Table 2), thus confirming that the use of spectral models can be considered a valid alternative for the determination of the optimal sample lenghts to retrieve $\epsilon$ from lidar data.

The use of spectral models to determine the appropriate sample size to use when retrieving $\epsilon$ from lidars can also be applied when information about atmospheric stability are not available or accurate. In these cases, instead of calculating an average optimal sample size for each stability condition, an appropriate time scale can be determined at each time $\epsilon$ is retrieved from lidar measurements, from a single spectrum. We compare $\epsilon$ values from the sonic anemometers and from the Halo Streamline lidar, with the optimal time scales obtained from both the proposed approaches (comparison with the sonic anemometer data and analysis of instantaneous spectra) in Figure 10, for the same time period shown in Figure 7. The use of spectral models to determine the extension of the inertial sub-range in the lidar spectra produces valid estimates of $\epsilon$: for this case we obtain a MAE= 0.40, and a correlation coefficient $R^2 = 0.78$.

**5   Variability of turbulence dissipation rate**

Once the capability of the method to provide accurate estimates of $\epsilon$ from lidar data has been tested, the variability of turbulence in the boundary layer can be assessed, using data from the various instruments deployed at XPIA.

The time series of $\epsilon$ shown in the previous  section revealed that, during the course of the day, $\epsilon$ changes by several orders of magnitude. To better explore this diurnal variability, Figure 11 shows the daily climatology of turbulence dissipation rate, calculated as median of the data from the sonic anemometer, WINDCUBE v2 lidar and Halo Streamline lidar. Plots for the two WINDCUBE v1 lidars are shown in the Supplementary Materials,  and are similar to the results from the WINDCUBE v2. A general good agreement between the climatology from sonic anemometers and lidars can be observed. A definite diurnal pattern is evident from each panel. As expected, the mainly quiescent conditions at night determine low

[Figure]

**Figure 10.** Time series from 6 April 2015 00 UTC to 10 April 2015 00 UTC comparing $\epsilon$ from sonic anemometers and the Halo Streamline lidars at 100m AGL, where the time scales for the lidars have been determined with both the proposed approaches (comparison with $\epsilon$ from sonic anemometers and fit with spectral models). Data have been smoothed with a 30-min running mean.

[Figure]

**Figure 11.** Daily climatology of turbulence dissipation rate derived from  raw values from the Halo Streamline (left), the WINDCUBE v2 lidar (center), and  sonic anemometers (right). Results from the two WINDCUBE v1s are included in the Supplementary Material.

values of turbulence dissipation rate ($\epsilon \sim 10^{-5} - 10^{-4} \, \mathrm{m}^2\mathrm{s}^{-3}$), while daytime convection increases the median turbulence dissipation in the boundary layer by several orders of magnitude ($\epsilon \sim 10^{-2} \, \mathrm{m}^2\mathrm{s}^{-3}$). During nighttime, however, the median values of $\epsilon$ show more variability than during daytime conditions, as traces of  intermittent bursts of $\epsilon$ can be detected in the climatology. We will investigate these changes in $\epsilon$ in more detail, by relating the

[revised manuscript text omitted]

have been observed several times (Lundquist et al., 2017). As case study, Figure 15 shows how wind speed, wind direction, and turbulence dissipation rate varied during the night between 6 - 7 April 2015, as measured by the Halo Streamline lidar. The analysis of the weather maps for this period reveals no frontal passage during the LLJ event, while a quasi-stationary front likely occurred at the end of the event ($\sim 23$ LT), as also confirmed by the  shift in wind direction during  this period, as shown in Figure 15b. No precipitation was recorded; and the analysis of ceilometer data reveals clear sky. A considerable increase in wind speed (up to $14\,\mathrm{m\,s^{-1}}$, Figure 15a) can be observed between 21 and 23 LT. In correspondence to this jet, turbulence dissipation rate (Figure 15c) increases by at least an order of magnitude throughout the considered vertical portion of the boundary layer,  as a consequence of an increase in wind speed variance, as observed in previous studies (Banta et al., 2006). $\epsilon$ reaches 
[revised manuscript text omitted]

Kristensen, L., Lenschow, D., Kirkegaard, P., and Courtney, M.: The spectral velocity tensor for homogeneous boundary-layer turbulence,

25  in: Boundary Layer Studies and Applications, pp. 149–193, Springer, 1989.

Lenschow, D., Mann, J., and Kristensen, L.: How long is long enough when measuring fluxes and other turbulence statistics?, Journal of Atmospheric and Oceanic Technology, 11, 661–673, 1994.

Lenschow, D. H., Wulfmeyer, V., and Senff, C.: Measuring second-through fourth-order moments in noisy data, Journal of Atmospheric and Oceanic Technology, 17, 1330–1347, 2000.

30  Lothon, M., Lenschow, D. H., and Mayor, S. D.: Doppler lidar measurements of vertical velocity spectra in the convective planetary boundary layer, Boundary-layer meteorology, 132, 205–226, 2009.

Lundquist, J. K. and Bariteau, L.: Dissipation of Turbulence in the Wake of a Wind Turbine, Boundary-Layer Meteorol, 154, 229–241, https://doi.org/10.1007/s10546-014-9978-3, http://link.springer.com/article/10.1007/s10546-014-9978-3, 2014.

Lundquist, J. K. and Chan, S. T.: Consequences of urban stability conditions for computational fluid dynamics simulations of urban disper-

35  sion, Journal of applied meteorology and climatology, 46, 1080–1097, 2007.

Lundquist, J. K., Wilczak, J. M., Ashton, R., Bianco, L., Brewer, W. A., Choukulkar, A., Clifton, A., Debnath, M., Delgado, R., Friedrich, K., et al.: Assessing state-of-the-art capabilities for probing the atmospheric boundary layer: the XPIA field campaign, Bulletin of the American Meteorological Society, 98, 289–314, 2017.

Mann, J.: The spatial structure of neutral atmospheric surface-layer turbulence, Journal of fluid mechanics, 273, 141–168, 1994.

McCaffrey, K., Bianco, L., and Wilczak, J. M.: Improved observations of turbulence dissipation rates from wind profiling radars, Atmospheric Measurement Techniques, 10, 2595–2611, https://doi.org/10.5194/amt-10-2595-2017, 2017a.

McCaffrey, K., Quelet, P. T., Choukulkar, A., Wilczak, J. M., Wolfe, D. E., Oncley, S. P., Brewer, W. A., Debnath, M., Ashton, R., Iungo, G. V., et al.: Identification of tower-wake distortions using sonic anemometer and lidar measurements, Atmospheric Measurement Techniques, 10, 393, 2017b.

Mirocha, J., Lundquist, J., and Kosović, B.: Implementation of a nonlinear subfilter turbulence stress model for large-eddy simulation in the Advanced Research WRF model, Monthly Weather Review, 138, 4212–4228, 2010.

Muñoz-Esparza, D., Sharman, R. D., and Lundquist, J. K.: Turbulent dissipation rate in the atmospheric boundary layer: observations and WRF mesoscale modeling during the XPIA field campaign, Monthly Weather Review, 2017.

Muñoz-Esparza, D., Cañadillas, B., Neumann, T., and van Beeck, J.: Turbulent fluxes, stability and shear in the offshore environment: Mesoscale modelling and field observations at FINO1, Journal of Renewable and Sustainable Energy, 4, 063 136, https://doi.org/10.1063/1.4769201, 2012.

Nakanishi, M. and Niino, H.: An improved Mellor–Yamada level-3 model: Its numerical stability and application to a regional prediction of advection fog, Boundary-Layer Meteorology, 119, 397–407, 2006.

Nakanishi, M. and Niino, H.: Development of an improved turbulence closure model for the atmospheric boundary layer, Journal of the Meteorological Society of Japan. Ser. II, 87, 895–912, 2009.

Newsom, R. K., Brewer, W. A., Wilczak, J. M., Wolfe, D. E., Oncley, S. P., and Lundquist, J. K.: Validating precision estimates in horizontal wind measurements from a Doppler lidar, Atmospheric Measurement Techniques, 10, 1229, 2017.

Nilsson, E., Lohou, F., Lothon, M., Pardyjak, E., Mahrt, L., and Darbieu, C.: Turbulence kinetic energy budget during the afternoon transition– Part 1: Observed surface TKE budget and boundary layer description for 10 intensive observation period days, Atmospheric Chemistry and Physics, 16, 8849–8872, 2016.

O'Connor, E. J., Illingworth, A. J., Brooks, I. M., Westbrook, C. D., Hogan, R. J., Davies, F., and Brooks, B. J.: A method for estimating the turbulent kinetic energy dissipation rate from a vertically pointing Doppler lidar, and independent evaluation from balloon-borne in situ measurements, Journal of Atmospheric and Oceanic Technology, 27, 1652–1664, 2010.

Olesen, H. R., Larsen, S. E., and Højstrup, J.: Modelling velocity spectra in the lower part of the planetary boundary layer, Boundary-Layer Meteorology, 29, 285–312, 1984.

Oncley, S. P., Friehe, C. A., Larue, J. C., Businger, J. A., Itsweire, E. C., and Chang, S. S.: Surface-layer fluxes, profiles, and turbulence measurements over uniform terrain under near-neutral conditions, Journal of the Atmospheric Sciences, 53, 1029–1044, 1996.

Panofsky, H. A.: Matching in the convective planetary boundary layer, Journal of the Atmospheric Sciences, 35, 272–276, 1978.

Paquin, J. and Pond, S.: The determination of the Kolmogoroff constants for velocity, temperature and humidity fluctuations from second-and third-order structure functions, Journal of Fluid Mechanics, 50, 257–269, 1971.

Pearson, G., Davies, F., and Collier, C.: An analysis of the performance of the UFAM pulsed Doppler lidar for observing the boundary layer, Journal of Atmospheric and Oceanic Technology, 26, 240–250, 2009.

Piper, M. and Lundquist, J. K.: Surface layer turbulence measurements during a frontal passage, Journal of the Atmospheric Sciences, 61, 1768–1780, 2004.

Prabha, T. V., Leclerc, M. Y., Karipot, A., and Hollinger, D. Y.: Low-frequency effects on eddy covariance fluxes under the influence of a low-level jet, Journal of Applied Meteorology and Climatology, 46, 338–352, 2007.

Rhodes, M. E. and Lundquist, J. K.: The Effect of Wind-Turbine Wakes on Summertime US Midwest Atmospheric Wind Profiles as Observed with Ground-Based Doppler Lidar, Boundary-Layer Meteorology, 149, 85–103, https://doi.org/10.1007/s10546-013-9834-x, 2013.

Rye, B.: Antenna parameters for incoherent backscatter heterodyne lidar, Applied Optics, 18, 1390–1398, 1979.

Skamarock, W. C., Klemp, J. B., Dudhia, J., Gill, D. O., Barker, D. M., Wang, W., and Powers, J. G.: A description of the advanced research
5    WRF version 2, Tech. rep., National Center For Atmospheric Research Boulder Co Mesoscale and Microscale Meteorology Div, 2005.

Smalikho, I.: On measurement of the dissipation rate of the turbulent energy with a cw Doppler lidar, ATMOSPHERIC AND OCEANIC OPTICS C/C OF OPTIKA ATMOSFERY I OKEANA, 8, 788–793, 1995.

Smalikho, I., Köpp, F., and Rahm, S.: Measurement of atmospheric turbulence by 2-$\mu$ m Doppler lidar, Journal of Atmospheric and Oceanic Technology, 22, 1733–1747, 2005.

10   Smalikho, I. N. and Banakh, V. A.: Measurements of wind turbulence parameters by a conically scanning coherent Doppler lidar in the atmospheric boundary layer, Atmospheric Measurement Techniques, 10, 4191, 2017.

Sobel, A. H. and Neelin, J. D.: The boundary layer contribution to intertropical convergence zones in the quasi-equilibrium tropical circulation model framework, Theoretical and Computational Fluid Dynamics, 20, 323–350, 2006.

Sreenivasan, K. R.: On the universality of the Kolmogorov constant, Physics of Fluids, 7, 2778–2784, 1995.

15   Taylor, G. I.: Statistical theory of turbulence, in: Proceedings of the Royal Society of London A: Mathematical, Physical and Engineering Sciences, vol. 151, pp. 421–444, The Royal Society, 1935.

Tennekes, H. and Lumley, J. L.: A first course in turbulence, MIT press, Cambridge, MA, 1972.

Tonttila, J., O'Connor, E., Hellsten, A., Hirsikko, A., O'Dowd, C., Järvinen, H., and Räisänen, P.: Turbulent structure and scaling of the inertial subrange in a stratocumulus-topped boundary layer observed by a Doppler lidar, Atmospheric chemistry and physics, 15, 5873–
20   5885, 2015.

Vanderwende, B. J., Lundquist, J. K., Rhodes, M. E., Takle, E. S., and Irvin, S. L.: Observing and simulating the summertime low-level jet in central Iowa, Monthly Weather Review, 143, 2319–2336, 2015.

Wang, H., Barthelmie, R. J., Doubrawa, P., and Pryor, S. C.: Errors in radial velocity variance from Doppler wind lidar, Atmospheric Measurement Techniques, 9, 4123, 2016.

25   Wharton, S. and Lundquist, J. K.: Assessing atmospheric stability and its impacts on rotor-disk wind characteristics at an onshore wind farm, Wind Energy, 15, 525–546, 2012.

Wilczak, J. M., Oncley, S. P., and Stage, S. A.: Sonic anemometer tilt correction algorithms, Boundary-Layer Meteorology, 99, 127–150, 2001.

Yang, B., Qian, Y., Berg, L. K., Ma, P.-L., Wharton, S., Bulaevskaya, V., Yan, H., Hou, Z., and Shaw, W. J.: Sensitivity of turbine-height wind
30   speeds to parameters in planetary boundary-layer and surface-layer schemes in the weather research and forecasting model, Boundary-Layer Meteorology, 162, 117–142, 2017.

Zhang, J. A., Drennan, W. M., Black, P. G., and French, J. R.: Turbulence structure of the hurricane boundary layer between the outer rainbands, Journal of the Atmospheric Sciences, 66, 2455–2467, 2009.

[Figure]

**Figure 15.** Nocturnal low-level jet case study. (a) Variability of wind speed from 20 UTC, 6 April 2015, to 8 UTC, 7 April 2015, as measured by the Halo Streamline lidar. (b) Wind direction at 116m AGL, during the same period of time. (c) Correspondent variability of turbulence dissipation rate $\epsilon$ as derived from the Halo Streamline measurements.

---

## Referee Report (RR1)

Thank you for your accommodations of my comments. The impact this manuscript can have is much increased with the addition of the new section, though in reality, the method of focus is likely to not be used without co-located sonic anemometers.

Comments on the new Section 4.1:
- Why the Kristensen model? Do you use isotropic or anisotropic, horizontally-homogenous?
- What do other models look like? That fit doesn't look great, but maybe that's the best there is?
- What kind of sensitivity/variability and error are introduced based on the choice of model and parameters? What kind of spread in time scales do you get?

Pages 5, line 5: what was the last date of data used?
Figure 6 caption: misspelling "variability"
Page 14, line 9: "likely contributes an additional error for this lidar"
Page 14 line 17: "the inherent uncertainty in the sonic anemometers' retrievals of \epsilon"
Page 17 line 5: spectral models can be fit
Add theoretical slope line to new Fig 9

---

## Author Response (AR2)

*In this document, the reviewers' comments are in black, the authors responses are in red.*

The authors thank the reviewers for their additional useful suggestions to improve the quality of our manuscript.

**Referee Report #1**

Thank you for your accommodations of my comments. The impact this manuscript can have is much increased with the addition of the new section, though in reality, the method of focus is likely to not be used without co-located sonic anemometers.

Comments on the new Section 4.1:
Why the Kristensen model? Do you use isotropic or anisotropic, horizontally-homogenous? What do other models look like? That fit doesn't look great, but maybe that's the best there is? What kind of sensitivity/variability and error are introduced based on the choice of model and parameters? What kind of spread in time scales do you get?

We agree with the reviewer that many different spectral models have been proposed in the literature. Many of these, however, have been developed assuming specific atmospheric stability conditions: Kaimal et al. (1972) developed a spectral model in the neutral limit; unstable conditions were assumed in the studies by Kaimal et al. (1976), Panosky (1978), Kaimal (1978), Højstrup (1981 and 1982); while Kaimal (1973) and Caughey (1977) proposed spectral models for stable conditions. The model proposed by Kristensen does not depend on specific stability conditions, which is a desirable condition considered that we are proposing the use of spectral models when co-located in-situ measurements (necessary to quantify atmospheric stability conditions) are not available. The Section in our manuscript includes several references to the papers where different spectral models have been proposed, and where the interested reader can find additional details. We have also included the following sentence to better clarify our choice: "We test the spectral model proposed by Kristeensen (1989), which proposes expressions for both the cases of an isotropic and an anisotropic horizontally homogeneous flow, without assumptions on the stability condition".
In addition, the Kristensen model (with the parameters we used) has been chosen in similar applications in recent studies – Lothon et al. 2009 and Tontilla et al. 2015, already included as references in the Section of our manuscript.
Moreover, we don't think that the specific choice of the functional form for the fit introduces a large uncertainty in our calculations. In fact, we think that the amount of information that a spectral model fit provides could even somehow be considered excessive, as our approach only aims at the determination of the extension of the inertial sub-range of the turbulence spectrum. As a consequence, we are also exploring the use of a simplified technique to determine the extension of the inertial sub-range by performing a local regression of the experimental spectra from the lidar, but we think that this further extension of the proposed method is beyond the scope of the present manuscript (where the lidars are co-located with a met tower, and so the calculation of spectra is not strictly necessary) and will be proposed in an additional manuscript in preparation.
Regarding the spread in the time scales we get, we think that a comprehensive analysis of these results for the proposed approach will better fit a later study, when the absence of co-located in situ measurements makes the use of lidar spectra the only available approach. In any case, for this

analysis, in unstable conditions the first quartile of the distribution of the timescales is 52s, while the third quartile is 83s. In stable conditions, the first quartile of the distribution of the timescales is 25s, while the third quartile is 43s. At this stage, we think it is sufficient to report in the Section of the present manuscript the values of MAE and correlation coefficient for the results presented here (Figure 10), as these are the metrics mostly used for all the comparisons made throughout the manuscript.

Finally, the chosen model assumes an anisotropic horizontally homogeneous flow. We have included the following sentence in the presentation of the model in the Section: " assuming an anisotropic horizontally homogeneous flow".

Pages 5, line 5: what was the last date of data used?
Data from the Halo lidar have been used until 16 April 2015 (as stated in the paragraph).

Figure 6 caption: misspelling "variability"
Corrected.

Page 14, line 9: "likely contributes an additional error for this lidar"
Corrected.

Page 14 line 17: "the inherent uncertainty in the sonic anemometers' retrievals of \epsilon"
Corrected.

Page 17 line 5: spectral models can be fit
Corrected.

Add theoretical slope line to new Fig 9
The line has been added to the plot and the caption changed accordingly.

**Referee Report #2**

The authors have clearly made significant changes to the manuscript in response to concerns initially raised, including the addition of another section to further the applicability of the technique and results presented here. All of my initial concerns have been sufficiently addressed, and I recommend that this article be accepted pending some technical corrections on the new section.

a) Eqs. 4 & 15: Modify these equations slightly to reflect the fact that the reported MAE values are %'s, as that is what is mostly used in the manuscript.
The equations now include a "· 100" term to show that the values are %'s.

b) Throughout manuscript: Please be consistent on whether MAE is being reported as a % (Table 2, Figure 6, most of manuscript) or as a fraction (p. 6 l. 25, Figure 5) so that the values are directly comparable with each other.
MAE is now always reported as %. Figure 5 has been changed accordingly.

c) Figure 9: Add units to the y-axis.
Added.

[revised manuscript text omitted]

is again necessary to retrieve the actual wind vector. These instruments will be identified in the remainder of the analysis with their serial numbers, 61 and 68.

Finally, a Halo Photonics Streamline Doppler scanning lidar (Pearson et al., 2009) from the U.S. Department of Energy Office of Science Atmospheric Radiation Measurement program was deployed from 6 March to 16 April 2015. This lidar used

5  a range gate resolution of 30m, with 200 total range gates. However, the maximum range gate with an acceptable number ($>30\%$) of valid measurements (SNR $>-20$dB) was at about 800m AGL. This scanning lidar was mainly used in a vertical staring mode. The scan strategy also included a 40-s plan-position-indicator (PPI) scan at an elevation angle of $60°$ once every 12min (from which the derivation of the horizontal wind speed is possible), a 10-min tower stare once per hour, and a target sector scan once per day to confirm heading relative to the tower (Newsom et al., 2017).

10  Table 1 includes the main technical characteristics of the three commercial lidar models considered in our analysis.

**3  Methods to estimate turbulence dissipation rate $\epsilon$**

**3.1  Turbulence dissipation from sonic anemometer**

Sonic anemometers data can be used to calculate turbulence dissipation rate with two different methods: the inertial sub-range energy spectra method and the second-order structure function method. Muñoz-Esparza et al. (2017) analyzed data at XPIA

15  and showed that the second-order structure function method has a lower error in estimating $\epsilon$ compared to the inertial sub-range energy spectra method, even when shorter overlapping temporal sub-windows are used to obtain a more regular pattern in the spectra. Therefore, we also apply the second-order structure function method to estimate $\epsilon$ from sonic anemometer measurements every 30s, for the 3-month period of XPIA.

According to Kolmogorov's hypothesis, within the inertial sub-range the velocity increments, expressed as second-order structure function $D_U$ of the horizontal velocity $U$, can be related to $\epsilon$ as:

$$D_U(r) \equiv\; <[U(x+r) - U(x)]^2> = \frac{1}{a}\epsilon^{2/3}r^{2/3} \tag{2}$$

where $< \cdot >$ denotes an ensemble average, and $a$ is the Kolmogorov constant. We assume $a = 0.52$, which is consistent with the range of values present in the literature (Paquin and Pond, 1971; Sreenivasan, 1995). The spatial separations $r$, which must be within the inertial sub-range, can be expressed as temporal velocity increments by invoking Taylor's frozen turbulence hypothesis (Taylor, 1935), so that $\epsilon$ can be determined as:

$$\epsilon = \frac{1}{U\tau}[aD_U(\tau)]^{3/2} \tag{3}$$

where $D_U(\tau)$ is the second-order structure function of the horizontal velocity $U$ calculated over temporal increments $\tau$. For every $\epsilon$ calculation (i.e. every $30$s), the second-order structure function was calculated with a 2-min window for $\tau$, centered at the nominal time at which $\epsilon$ is calculated. Then, the fitting to the theoretical model only used the time range between $\tau = 0.1$s and $\tau = 2$s. Such a short temporal separation in the data is expected to lie well within the inertial sub-range, therefore excluding the undesired contributions from the outer scales which would undermine Kolmogorov's fundamental assumptions. Moreover, despite the reduced size of the chosen time range, the high temporal resolution of the sonic anemometers still guarantees an adequate number of data points to allow a robust estimation of the structure function. Data inspection confirms that the desired theoretical $\tau^{2/3}$ slope is observed in the chosen range for $\tau$ (example shown in the Supplement).

As already mentioned, data were excluded for wind directions waked by the tower. When neither of the two anemometers is affected by tower wakes, $\epsilon$ is defined as the average between the two independent values obtained from the two sonics at each height. To quantify the uncertainty in turbulence dissipation rate measurements from the sonic anemometers, we have compared $\epsilon$ from the two sonics at each level when neither one was influenced by the tower wake. For each tower boom direction (northwest and southeast), we calculate the median absolute error (MAE) between $\epsilon$ from the sonic anemometers mounted on the considered boom direction and the correspondent average value from the two sonics:

$$\mathrm{MAE} = \mathrm{median}\left(\frac{|\epsilon_{single} - \epsilon_{average}|}{\epsilon_{average}}\right) \cdot 100 \tag{4}$$

In calculating the error, we consider data from all heights, as no significant difference was noticed at different levels. For both the boom directions, we find very similar results, with  MAE = 19%, which is reduced to  14% when a 30-min running mean is applied to the $\epsilon$ time series. The distributions of the errors are included in the Supplementary Material. No bias was detected between the retrievals from the sonic anemometers on the two boom directions.

**3.2 Dissipation from Doppler lidar**

Wind Doppler lidars can provide a great improvement of our understanding of the variability of turbulence dissipation thanks to the ease of their deployment in different locations and the long measurement range allowed by several commercial models. To do so, robust methods to estimate $\epsilon$ with lidars are necessary, and their uncertainty has to be assessed. For this purpose, we

[Figure]

**Figure 2.** Turbulence energy spectrum according to Kolmogorov's hypothesis.

follow and refine the approach described in O'Connor et al. (2010) to estimate $\epsilon$ from vertical profiling lidars or scanning lidars used in vertical staring mode. For homogeneous and isotropic turbulence, within the inertial sub-range, the turbulent energy spectrum (Figure 2) can be expressed according to the Kolmogorov (1941) hypothesis in terms of wavenumber $k$ as:

$$S(k) = a\epsilon^{2/3}k^{-5/3} \tag{5}$$

5    where $a \simeq 0.52$ is the one-dimensional Kolmogorov constant. The wavenumber $k$ can be written in terms of a length scale $L = 2\pi/k$ by invoking Taylor's frozen turbulence hypothesis (Taylor, 1935). By integrating (5) over the wavenumber space, starting from the wavenumber $k_1$ correspondent to a single lidar sample, the variance $\sigma_v^2$ of the de-trended observed line-of-sight velocity from $N$ samples can be obtained:

$$\sigma_v^2 = \int_k^{k_1} S(k)dk = -\frac{3}{2}a\epsilon^{2/3}\left(k_1^{-2/3} - k^{-2/3}\right) = \tag{6}$$

$$= \frac{3a}{2}\left(\frac{\epsilon}{2\pi}\right)^{2/3}\left(L_N^{2/3} - L_1^{2/3}\right) \tag{7}$$

and therefore if the length scales are properly chosen (and consistent with how $\sigma_v$ is computed) then $\epsilon$ can be calculated without the need of systematically computing turbulence energy spectra. In (7), the length scale $L_1$ for a single sample interval is given by:

$$L_1 = Ut + 2z\sin\left(\frac{\theta}{2}\right) \tag{8}$$

15    where $U$ is the horizontal wind speed, $t$ is the dwell time, $\theta$ the half-angle divergence of the lidar beam, and $z$ the height AGL. Since Doppler lidars generally have a very small $\theta$ ($< 0.1\,\mathrm{mrad}$), the second term in (8) is typically negligible. For $N$ samples, the length scale becomes $L_N = NUt$. For the WINDCUBE lidars, the variance of the observed line-of-sight velocity $\sigma_v^2$ can be calculated as average from all the beams. In doing so, we include turbulence contributions from both the horizontal and vertical dimensions, and we make the limiting (Kaimal et al., 1972; Mann, 1994) assumption of isotropic turbulence. For

the Halo Streamline lidar, which operated in a vertical stare mode, $\sigma_v^2$ is calculated from the vertically pointing beam, and therefore $\epsilon$ will strictly include turbulence contributions only in the vertical dimension, thus possibly determining different values compared to what is retrieved from the WINDCUBE lidars. Another difference due to the different scan patterns used by the considered lidars is related to the determination of the horizontal wind speed $U$. For the WINDCUBE lidars, $U$ can be derived from the line-of-sight velocity measurements from the different beams, with the assumption of horizontal homogeneity of the flow over the probed volume. In the case of the Halo Streamline, no information about the horizontal wind can be derived from the measurements in the vertical staring mode, which only measures the vertical component of the wind speed. $U$ is then retrieved from a sine-wave fitting from the VAD scans that are performed every $12\,\mathrm{min}$. The heights at which the measurements are taken during the tilted VAD scans are not the same as the heights sampled in the vertical staring mode. Therefore, for each considered level in the vertical staring mode, $U$ is determined from a linear interpolation of the wind speed retrieved at the two closest heights during the VAD scans. Considerations about the error introduced by this procedure on the estimation of $\epsilon$ will be discussed in Section 4.

Lidar measurements are inherently affected by signal noise as well as possible variations of the aerosol fall speeds, which provide additional contributions to the observed variance. By assuming that the contribution of all atmospheric flows to the observed line-of-sight variance within the considered short time scales can be regarded as of turbulent nature, 
[revised manuscript text omitted]

[Figure]

**Figure 5.** Median absolute error between $\epsilon$ estimates (smoothed with a 30-min running mean) from sonic anemometer and WINDCUBE v2 lidar data at 100m AGL during the whole period of the XPIA campaign, as a function of the sample length used to estimate $\epsilon$ from lidar data.

scales for different stability conditions, consistency with the time scale used to calculate turbulent fluxes for the determination of the Obukhov Length $L$ is advisable. Therefore, a 30-min running mean is applied to the time series of $\epsilon$ from both sonic anemometers and lidars before comparing the estimates from the different instruments.

To quantify the difference between sonic and lidar estimates of $\epsilon$, we use the median absolute error (MAE), defined as:

$$\quad \mathrm{MAE} = \mathrm{median}\left(\frac{|\epsilon_{lidar} - \epsilon_{sonic}|}{\epsilon_{sonic}}\right) \cdot 100 \tag{15}$$

[revised manuscript text omitted]

**4.1 Determination of the optimal time scales to retrieve $\epsilon$ from lidars in absence of co-located sonic anemometers**

5   The availability of multiple sonic anemometers co-located with the lidars at XPIA has allowed for a direct comparison between $\epsilon$ estimates from different instruments to determine the optimal length scales, in different stability conditions, to use when retrieving $\epsilon$ from Doppler lidar measurements. This approach does not require the direct calculation of spectra from the line-of-sight velocity measured by the lidars, and therefore it represents a time-efficient technique. However, the proposed method is only viable when sonic anemometers are deployed in the near vicinity of a lidar, and when measures of atmospheric stability

10   are available.

When a comparison with sonic anemometer data is not possible, the appropriate time scale to use in the lidar retrieval of $\epsilon$ can be determined by finding the maximum wavelength within the inertial sub-range in the velocity spectra from the lidar measurements. To do so, spectral models can be  fit to the observed spectra. Several models have been proposed for turbulence spectra in different stability conditions (Kaimal et al., 1972; Panofsky, 1978; Olesen et al., 1984). We test the

15   spectral model proposed by Kristensen et al. (1989), which proposes expressions for both the cases of an isotropic and an anisotropic horizontally homogeneous flow, without assumptions on the stability condition. To validate our results and test this alternative approach to derive $\epsilon$ from lidar measurements, we use data from the Halo Streamline lidar to estimate the maximum wavelength $\lambda_z$ within the inertial subrange. Since the Halo mainly operated in a vertical stare mode during XPIA, we consider the following expression for the turbulence spectrum of the vertical component of the wind speed, assuming an anisotropic

20   horizontally homogeneous flow:

$$S(k) = \frac{\sigma_z^2 l_z}{2\pi} \frac{1 + \frac{8}{3}\left(\frac{l_z k}{a(\mu)}\right)^{2\mu}}{\left[1 + \left(\frac{l_z k}{a(\mu)}\right)^{2\mu}\right]^{5/(6\mu)+1}} \tag{16}$$

where $k$ is the wavenumber, $\sigma_z$ is the standard deviation of the vertical component of the wind speed used to compute the spectrum, $l_z$ is the integral scale of the vertical velocity along the horizontal flow trajectory, and the parameter $\mu$ controls the curvature of the spectrum. We use $\mu = 1.5$, which provides a good match with our experimental spectra, as also found in

25   previous studies (Lothon et al., 2009; Tonttila et al., 2015). The parameter $a$ can be expressed as a function of $\mu$ as:

$$a(\mu) = \pi \frac{\mu \Gamma\left(\frac{5}{6\mu}\right)}{\Gamma\left(\frac{1}{2\mu}\right)\Gamma\left(\frac{1}{3\mu}\right)} \tag{17}$$

We calculate spectra using 10-min consecutive data, and we fit the spectral model to the experimental data, leaving out frequencies greater than $0.2$Hz, which are affected by instrumental noise (Frehlich, 2001), not modeled here. An example of a measured spectrum and the fit resulting from the model are shown in Figure 9. The transition wavelength $\lambda_z$ between the

[Figure]

**Figure 9.** Example of power spectral density of the vertical component of the wind speed as measured by the Halo Streamline lidar on 11 March 2015 18:05 UTC. The red line represents the fit according to the spectral model from Eq. (16), the orange dotted line shows the theoretical slope.

inertial sub-range and the outer scales can be expressed as a function of the integral scale $l_z$ and the parameter $\mu$:

$$\lambda_z = \left[ \frac{5}{3}\sqrt{\mu^2 + \frac{6}{5}\mu + 1} - \left( \frac{5}{3}\mu + 1 \right) \right]^{1/(2\mu)} \frac{2\pi}{a(\mu)} l_z \tag{18}$$

Following the approach in Tonttila et al. (2015), we estimate the timescale corresponding to this transition wavelength by dividing $\lambda_z$ by the collocated wind speed derived from the closest PPI scan performed by the Halo Streamline lidar.

To compare the results from this approach with what we obtain from the comparison with dissipation rates from the sonic anemometer data, we apply this technique to the data from the Halo Streamline for the whole period of XPIA, and calculate the average timescales for different stability conditions at $100\mathrm{m}$ AGL. We obtain an average time scale of $32\mathrm{s}$ in stable conditions, and $73\mathrm{s}$ in unstable conditions. Both these values compare well with what is found with the more time-efficient comparison with the sonic anemometer retrievals (values in Table 2), thus confirming that the use of spectral models can be considered a valid alternative for the determination of the optimal sample lenghts to retrieve $\epsilon$ from lidar data.

The use of spectral models to determine the appropriate sample size to use when retrieving $\epsilon$ from lidars can also be applied when information about atmospheric stability are not available or accurate. In these cases, instead of calculating an average optimal sample size for each stability condition, an appropriate time scale can be determined at each time $\epsilon$ is retrieved from lidar measurements, from a single spectrum. We compare $\epsilon$ values from the sonic anemometers and from the Halo Streamline lidar, with the optimal time scales obtained from both the proposed approaches (comparison with the sonic anemometer data and analysis of instantaneous spectra) in Figure 10, for the same time period shown in Figure 7. The use of spectral models to determine the extension of the inertial sub-range in the lidar spectra produces valid estimates of $\epsilon$: for this case we obtain a MAE  $= 40\%$, and a correlation coefficient $R^2 = 0.78$.

[Figure]

**Figure 10.** Time series from 6 April 2015 00 UTC to 10 April 2015 00 UTC comparing $\epsilon$ from sonic anemometers and the Halo Streamline lidars at 100m AGL, where the time scales for the lidars have been determined with both the proposed approaches (comparison with $\epsilon$ from sonic anemometers and fit with spectral models). Data have been smoothed with a 30-min running mean.

**5    Variability of turbulence dissipation rate**

Once the capability of the method to provide accurate estimates of $\epsilon$ from lidar data has been tested, the variability of turbulence in the boundary layer can be assessed, using data from the various instruments deployed at XPIA.

5    The time series of $\epsilon$ shown in the previous section revealed that, during the course of the day, $\epsilon$ changes by several orders of magnitude. To better explore this diurnal variability, Figure 11 shows the daily climatology of turbulence dissipation rate, calculated as median of the data from the sonic anemometer, WINDCUBE v2 lidar and Halo Streamline lidar. Plots for the two WINDCUBE v1 lidars are shown in the Supplementary Materials, and are similar to the results from the WINDCUBE v2. A general good agreement between the climatology from sonic anemometers and lidars can be observed. A definite diurnal pattern is evident from each panel. As expected, the mainly quiescent conditions at night determine low values of turbulence dissipation

10   rate ($\epsilon \sim 10^{-5} - 10^{-4}\,\mathrm{m^2 s^{-3}}$), while daytime convection increases the median turbulence dissipation in the boundary layer by several orders of magnitude ($\epsilon \sim 10^{-2}\,\mathrm{m^2 s^{-3}}$). During nighttime, however, the median values of $\epsilon$ show more variability than during daytime conditions, as traces of intermittent bursts of $\epsilon$ can be detected in the climatology. We will investigate these changes in $\epsilon$ in more detail, by relating the variability of $\epsilon$ with wind speed, especially in the case of nocturnal low-level jets.

Also, the study of the climatology of $\epsilon$ can give insights on how $\epsilon$ changes with height. The analysis of the climatology from

15   the sonic anemometers (right panel in Figure 11), which allow measurements of $\epsilon$ at 5m AGL, shows how $\epsilon$ is higher close to the surface throughout the day, while above 50m AGL the change of $\epsilon$ with height is less noticeable. A similar result can be found from lidars, which provide $\epsilon$ measurements starting at 40m AGL for the WINDCUBE v2, and 75m AGL for the Halo

[Figure]

**Figure 11.** Daily climatology of turbulence dissipation rate derived from raw values from the Halo Streamline (left), the WINDCUBE v2 lidar (center), and sonic anemometers (right). Results from the two WINDCUBE v1s are included in the Supplementary Material.

Streamline, with reduced variability of $\epsilon$ with height in the majority of the sampled height range. The slight increase of $\epsilon$ above $\sim 600$m AGL at night for the Halo Streamline lidar (left panel in Figure 11) can be explained as due to more random errors in the line-of-sight velocity measured by the lidar at high altitudes but also as effect of the higher frequency of good-quality measurements at higher levels during high wind speed events, which determine higher turbulence, as will be shown later in this

5    section. A systematic analysis of how turbulence dissipation rate varies with height is shown in Figure 12. For each instrument, the percentage difference in $\epsilon$ is shown, and it is calculated by taking as reference value the $\epsilon$ value closest in time from the sonic anemometer at 5m AGL, so that a common reference level is identified for all the instruments. The continuous line in the plot shows the median value at each height, while the shaded band represents the 1st and 3rd quartiles of the data distribution. The plot confirms that turbulence dissipation rate shows most of its variability with height close to the surface, as also found

10    by Balsley et al. (2006). A 75% decrease in the median $\epsilon$ value is observed moving from 5m AGL to 50m AGL for the sonic anemometer data. We expect this large reduction in $\epsilon$ to be due to a rapid decrease in shear production with height close to the surface, as it has been shown (Nilsson et al., 2016) that shear production has a strong connection with dissipation close to the surface. An additional increase of height determines a lower rate of average reduction of $\epsilon$ with height, with the median $\epsilon$ values for the sonics experiencing an additional 15% reduction (compared to the reference 5m AGL level) between 50m AGL

[revised manuscript text omitted]

10    at the end of the event ($\sim$ 23 LT), as also confirmed by the shift in wind direction during this period, as shown in Figure 15b. No precipitation was recorded; and the analysis of ceilometer data reveals clear sky. A considerable increase in wind speed (up to $14\,\mathrm{m\,s}^{-1}$, Figure 15a) can be observed between 21 and 23 LT. In correspondence to this jet, turbulence dissipation rate (Figure 15c) increases by at least an order of magnitude throughout the considered vertical portion of the boundary layer, as a consequence of an increase in wind speed variance, as observed in previous studies (Banta et al., 2006). $\epsilon$ reaches values of

[revised manuscript text omitted]

Kristensen, L., Lenschow, D., Kirkegaard, P., and Courtney, M.: The spectral velocity tensor for homogeneous boundary-layer turbulence, in: Boundary Layer Studies and Applications, pp. 149–193, Springer, 1989.

Lenschow, D., Mann, J., and Kristensen, L.: How long is long enough when measuring fluxes and other turbulence statistics?, Journal of Atmospheric and Oceanic Technology, 11, 661–673, 1994.

Lenschow, D. H., Wulfmeyer, V., and Senff, C.: Measuring second-through fourth-order moments in noisy data, Journal of Atmospheric and Oceanic Technology, 17, 1330–1347, 2000.

30 Lothon, M., Lenschow, D. H., and Mayor, S. D.: Doppler lidar measurements of vertical velocity spectra in the convective planetary boundary layer, Boundary-layer meteorology, 132, 205–226, 2009.

Lundquist, J. K. and Bariteau, L.: Dissipation of Turbulence in the Wake of a Wind Turbine, Boundary-Layer Meteorol, 154, 229–241, https://doi.org/10.1007/s10546-014-9978-3, http://link.springer.com/article/10.1007/s10546-014-9978-3, 2014.

Lundquist, J. K. and Chan, S. T.: Consequences of urban stability conditions for computational fluid dynamics simulations of urban dispersion, Journal of applied meteorology and climatology, 46, 1080–1097, 2007.

35 Lundquist, J. K., Wilczak, J. M., Ashton, R., Bianco, L., Brewer, W. A., Choukulkar, A., Clifton, A., Debnath, M., Delgado, R., Friedrich, K., et al.: Assessing state-of-the-art capabilities for probing the atmospheric boundary layer: the XPIA field campaign, Bulletin of the American Meteorological Society, 98, 289–314, 2017.

Mann, J.: The spatial structure of neutral atmospheric surface-layer turbulence, Journal of fluid mechanics, 273, 141–168, 1994.

McCaffrey, K., Bianco, L., and Wilczak, J. M.: Improved observations of turbulence dissipation rates from wind profiling radars, Atmospheric Measurement Techniques, 10, 2595–2611, https://doi.org/10.5194/amt-10-2595-2017, 2017a.

McCaffrey, K., Quelet, P. T., Choukulkar, A., Wilczak, J. M., Wolfe, D. E., Oncley, S. P., Brewer, W. A., Debnath, M., Ashton, R., Iungo, G. V., et al.: Identification of tower-wake distortions using sonic anemometer and lidar measurements, Atmospheric Measurement Techniques, 10, 393, 2017b.

Mirocha, J., Lundquist, J., and Kosović, B.: Implementation of a nonlinear subfilter turbulence stress model for large-eddy simulation in the Advanced Research WRF model, Monthly Weather Review, 138, 4212–4228, 2010.

Muñoz-Esparza, D., Sharman, R. D., and Lundquist, J. K.: Turbulent dissipation rate in the atmospheric boundary layer: observations and WRF mesoscale modeling during the XPIA field campaign, Monthly Weather Review, 2017.

Muñoz-Esparza, D., Cañadillas, B., Neumann, T., and van Beeck, J.: Turbulent fluxes, stability and shear in the offshore environment: Mesoscale modelling and field observations at FINO1, Journal of Renewable and Sustainable Energy, 4, 063 136, https://doi.org/10.1063/1.4769201, 2012.

Nakanishi, M. and Niino, H.: An improved Mellor–Yamada level-3 model: Its numerical stability and application to a regional prediction of advection fog, Boundary-Layer Meteorology, 119, 397–407, 2006.

Nakanishi, M. and Niino, H.: Development of an improved turbulence closure model for the atmospheric boundary layer, Journal of the Meteorological Society of Japan. Ser. II, 87, 895–912, 2009.

Newsom, R. K., Brewer, W. A., Wilczak, J. M., Wolfe, D. E., Oncley, S. P., and Lundquist, J. K.: Validating precision estimates in horizontal wind measurements from a Doppler lidar, Atmospheric Measurement Techniques, 10, 1229, 2017.

Nilsson, E., Lohou, F., Lothon, M., Pardyjak, E., Mahrt, L., and Darbieu, C.: Turbulence kinetic energy budget during the afternoon transition–Part 1: Observed surface TKE budget and boundary layer description for 10 intensive observation period days, Atmospheric Chemistry and Physics, 16, 8849–8872, 2016.

O'Connor, E. J., Illingworth, A. J., Brooks, I. M., Westbrook, C. D., Hogan, R. J., Davies, F., and Brooks, B. J.: A method for estimating the turbulent kinetic energy dissipation rate from a vertically pointing Doppler lidar, and independent evaluation from balloon-borne in situ measurements, Journal of Atmospheric and Oceanic Technology, 27, 1652–1664, 2010.

Olesen, H. R., Larsen, S. E., and Højstrup, J.: Modelling velocity spectra in the lower part of the planetary boundary layer, Boundary-Layer Meteorology, 29, 285–312, 1984.

Oncley, S. P., Friehe, C. A., Larue, J. C., Businger, J. A., Itsweire, E. C., and Chang, S. S.: Surface-layer fluxes, profiles, and turbulence measurements over uniform terrain under near-neutral conditions, Journal of the Atmospheric Sciences, 53, 1029–1044, 1996.

Panofsky, H. A.: Matching in the convective planetary boundary layer, Journal of the Atmospheric Sciences, 35, 272–276, 1978.

Paquin, J. and Pond, S.: The determination of the Kolmogoroff constants for velocity, temperature and humidity fluctuations from second-and third-order structure functions, Journal of Fluid Mechanics, 50, 257–269, 1971.

Pearson, G., Davies, F., and Collier, C.: An analysis of the performance of the UFAM pulsed Doppler lidar for observing the boundary layer, Journal of Atmospheric and Oceanic Technology, 26, 240–250, 2009.

Piper, M. and Lundquist, J. K.: Surface layer turbulence measurements during a frontal passage, Journal of the Atmospheric Sciences, 61, 1768–1780, 2004.

Prabha, T. V., Leclerc, M. Y., Karipot, A., and Hollinger, D. Y.: Low-frequency effects on eddy covariance fluxes under the influence of a low-level jet, Journal of Applied Meteorology and Climatology, 46, 338–352, 2007.

Rhodes, M. E. and Lundquist, J. K.: The Effect of Wind-Turbine Wakes on Summertime US Midwest Atmospheric Wind Profiles as Observed with Ground-Based Doppler Lidar, Boundary-Layer Meteorology, 149, 85–103, https://doi.org/10.1007/s10546-013-9834-x, 2013.

Rye, B.: Antenna parameters for incoherent backscatter heterodyne lidar, Applied Optics, 18, 1390–1398, 1979.

Skamarock, W. C., Klemp, J. B., Dudhia, J., Gill, D. O., Barker, D. M., Wang, W., and Powers, J. G.: A description of the advanced research
5    WRF version 2, Tech. rep., National Center For Atmospheric Research Boulder Co Mesoscale and Microscale Meteorology Div, 2005.

Smalikho, I.: On measurement of the dissipation rate of the turbulent energy with a cw Doppler lidar, ATMOSPHERIC AND OCEANIC OPTICS C/C OF OPTIKA ATMOSFERY I OKEANA, 8, 788–793, 1995.

Smalikho, I., Köpp, F., and Rahm, S.: Measurement of atmospheric turbulence by 2-$\mu$ m Doppler lidar, Journal of Atmospheric and Oceanic Technology, 22, 1733–1747, 2005.

10   Smalikho, I. N. and Banakh, V. A.: Measurements of wind turbulence parameters by a conically scanning coherent Doppler lidar in the atmospheric boundary layer, Atmospheric Measurement Techniques, 10, 4191, 2017.

Sobel, A. H. and Neelin, J. D.: The boundary layer contribution to intertropical convergence zones in the quasi-equilibrium tropical circulation model framework, Theoretical and Computational Fluid Dynamics, 20, 323–350, 2006.

Sreenivasan, K. R.: On the universality of the Kolmogorov constant, Physics of Fluids, 7, 2778–2784, 1995.

15   Taylor, G. I.: Statistical theory of turbulence, in: Proceedings of the Royal Society of London A: Mathematical, Physical and Engineering Sciences, vol. 151, pp. 421–444, The Royal Society, 1935.

Tennekes, H. and Lumley, J. L.: A first course in turbulence, MIT press, Cambridge, MA, 1972.

Tonttila, J., O'Connor, E., Hellsten, A., Hirsikko, A., O'Dowd, C., Järvinen, H., and Räisänen, P.: Turbulent structure and scaling of the inertial subrange in a stratocumulus-topped boundary layer observed by a Doppler lidar, Atmospheric chemistry and physics, 15, 5873–
20   5885, 2015.

Vanderwende, B. J., Lundquist, J. K., Rhodes, M. E., Takle, E. S., and Irvin, S. L.: Observing and simulating the summertime low-level jet in central Iowa, Monthly Weather Review, 143, 2319–2336, 2015.

Wang, H., Barthelmie, R. J., Doubrawa, P., and Pryor, S. C.: Errors in radial velocity variance from Doppler wind lidar, Atmospheric Measurement Techniques, 9, 4123, 2016.

25   Wharton, S. and Lundquist, J. K.: Assessing atmospheric stability and its impacts on rotor-disk wind characteristics at an onshore wind farm, Wind Energy, 15, 525–546, 2012.

Wilczak, J. M., Oncley, S. P., and Stage, S. A.: Sonic anemometer tilt correction algorithms, Boundary-Layer Meteorology, 99, 127–150, 2001.

Yang, B., Qian, Y., Berg, L. K., Ma, P.-L., Wharton, S., Bulaevskaya, V., Yan, H., Hou, Z., and Shaw, W. J.: Sensitivity of turbine-height wind
30   speeds to parameters in planetary boundary-layer and surface-layer schemes in the weather research and forecasting model, Boundary-Layer Meteorology, 162, 117–142, 2017.

Zhang, J. A., Drennan, W. M., Black, P. G., and French, J. R.: Turbulence structure of the hurricane boundary layer between the outer rainbands, Journal of the Atmospheric Sciences, 66, 2455–2467, 2009.

[Figure]

**Figure 15.** Nocturnal low-level jet case study. (a) Variability of wind speed from 20 UTC, 6 April 2015, to 8 UTC, 7 April 2015, as measured by the Halo Streamline lidar. (b) Wind direction at 116m AGL, during the same period of time. (c) Correspondent variability of turbulence dissipation rate $\epsilon$ as derived from the Halo Streamline measurements.